# Loss of sea ice alters light spectra for aquatic photosynthesis

Monika Soja-Woźniak [1] ✉, Tadzio Holtrop[1,2], Sander Woutersen [3], Hendrik Jan van der Woerd[2], Lars Chresten Lund-Hansen[4] & Jef Huisman [1] ✉

The dramatic loss of sea ice due to global warming is changing light conditions for marine primary production, but exactly how is not well understood. Previous studies revealed that small peaks in the absorption spectrum of liquid water, due to molecular vibrations of $H_2O$, delineate a series of spectral niches for aquatic photosynthesis. Ice, however, has a smoother absorption spectrum and scatters light much more strongly than liquid water. Here, we show with a radiative transfer model that the loss of sea ice causes a pronounced blue shift, narrowing light spectra in the euphotic zone to shorter wavelengths. Furthermore, ice cover yields a smooth continuum of light spectra, whereas open water creates distinct spectral niches selecting for phytoplankton species with different photosynthetic pigments. These results indicate that the loss of sea ice will cause major changes in both the pigment and species composition of primary producers in polar ecosystems.

Although sea ice still covers almost 10% of the Earth's surface, in many areas the ice cover rapidly diminishes. The Arctic Ocean's ice cover has decreased dramatically, and predictions suggest an ice-free Arctic Ocean in summer within the next few decades[1,2]. The Antarctic sea ice extent increased slightly from the 1980s to 2015, but this trend reversed to a major decline since 2016[3]. Both sea ice and open ocean waters are inhabited by numerous photosynthetic microorganisms, including ice algae and phytoplankton, whose primary production provides the basis for the marine food web[4–6]. The high light intensities in the surface layer to which these primary producers will be exposed when the ice melts can have major effects on photoacclimation and community composition, and the implications of increased irradiance have therefore been extensively investigated[7–9]. However, laboratory experiments, field studies, and models have shown that the spectral composition of light is also a major selective factor among aquatic primary producers[10–16]. Hence, to what extent will the loss of sea ice alter the spectral composition of light available for aquatic photosynthesis in these new ice-free regions of the oceans?

Ice has a much higher albedo than liquid water, reflecting 20–90% of the incident solar radiation, depending on the presence of e.g., melt ponds or snow cover at the ice surface[17–19]. Furthermore, scattering of light in liquid water is mainly governed by $H_2O$ molecules and small suspended particles, is highest in the blue part of the spectrum, and declines at longer wavelengths. In contrast, scattering by sea ice is dominated by air bubbles and brine inclusions embedded in the ice[20], which strongly scatter light in an essentially wavelength-independent manner[21]. Hence, although the amount of scattering varies among ice types[17], sea ice generally has a much higher scattering coefficient and its scattering spectrum tends to be more "white" than that of liquid water (Fig. 1a).

At first glance, light absorption by ice and liquid water appears quite similar, with low absorption in the blue and high absorption in the red and infrared part of the spectrum (Fig. 1b). In liquid water, $H_2O$ molecules are constantly moving, while the covalent bonds between their oxygen and hydrogen atoms display three types of vibrations: symmetric stretching, antisymmetric stretching and bending. The overtones (harmonics) of these vibrational modes create a series of small peaks in the visible and near-infrared range of the absorption spectrum of liquid water[22–25]. Specifically, the small peaks at 401, 449, 514, 605 and 742 nm correspond with the 8th, 7th,

[1]Department of Freshwater and Marine Ecology, Institute for Biodiversity and Ecosystem Dynamics, University of Amsterdam, Amsterdam, the Netherlands. [2]Department of Water & Climate Risk, Institute for Environmental Studies (IVM), VU University Amsterdam, Amsterdam, the Netherlands. [3]Van't Hoff Institute for Molecular Sciences, University of Amsterdam, Amsterdam, The Netherlands. [4]Department of Biology, Arctic Research Centre, Aquatic Biology, Aarhus University, Aarhus, Denmark. ✉e-mail: monika@soja-wozniak.com; j.huisman@uva.nl

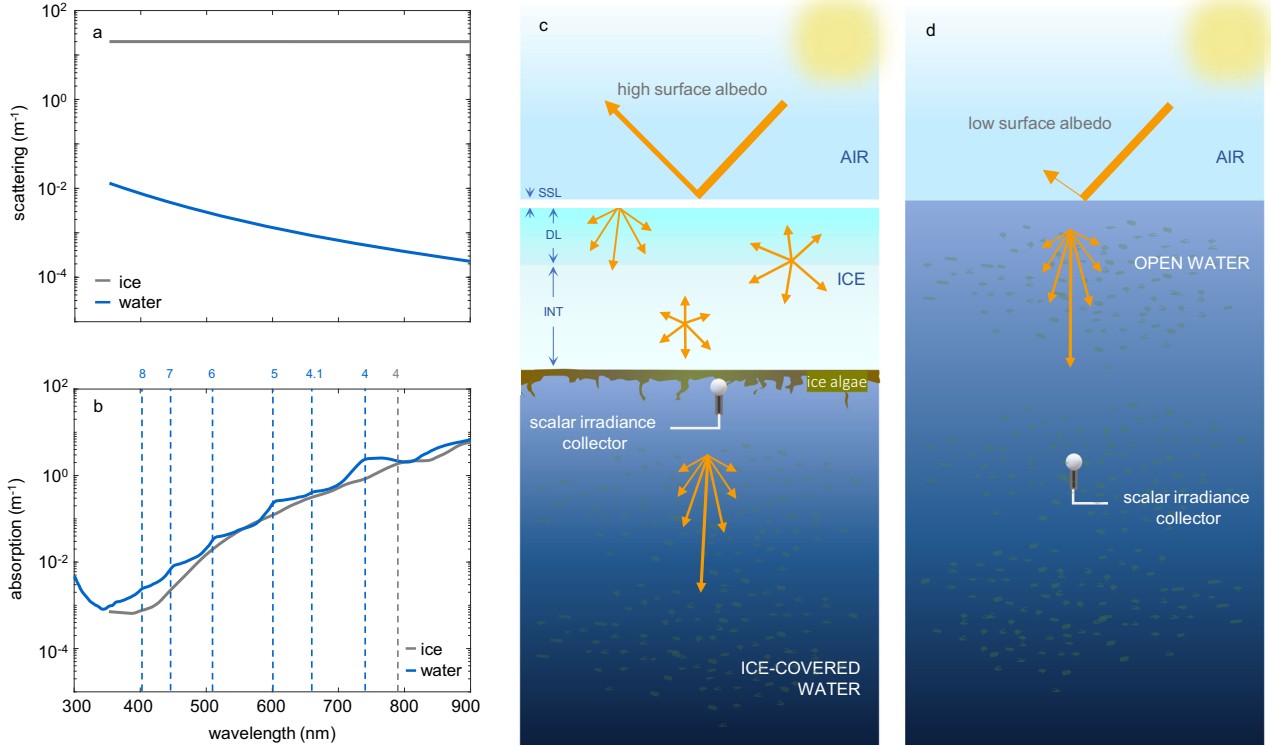

**Fig. 1 | Sea ice and seawater differ in their absorption and scattering properties. a** Light scattering spectra of sea ice (gray line) and seawater[22] (blue line). The scattering coefficient of sea ice ($b_{ice}$) varies depending on the amount of air bubbles and brine inclusions. Here we show a value of $b_{ice} = 20\,m^{-1}$, representative for the interior layer of Arctic sea ice[31]. **b** Light absorption spectra of pure ice[71] (solid gray line) and pure liquid water[23,25] (solid blue line). Vertical dashed lines indicate the harmonics of the vibrational modes of $H_2O$ molecules, for liquid water (blue) and ice (gray). **c, d** The two scenarios compared in this paper: irradiance spectra below ice cover versus in open water; note that ice-covered areas have a much higher albedo and show stronger scattering than open water. We assume that the ice cover consists of three layers: SSL surface scattering layer, DL drained layer, IL interior layer.

6th, 5th and 4th harmonics of the symmetric and antisymmetric O-H stretching mode[23,26,27] (Fig. 1b). Furthermore, the subtle peak at 662 nm corresponds with a combination of O-H stretching and bending modes known as the 4.1 sub-harmonic. It has been shown[11,15] that these small peaks in the absorption spectrum create large gaps in underwater light spectra, because light is attenuated exponentially with depth. These gaps delineate a series of distinct 'spectral niches' in aquatic ecosystems[11,15,28]. For example, light absorption by the 6th harmonics at 514 nm separates the blue niche (449–514 nm) of clear ocean waters from the green niche (514–605 nm) of many coastal waters. Likewise, light absorption by the 5th harmonics at 605 nm separates the green niche of coastal waters from the orange (605–662 nm) and red niches (662–742 nm) of humic lakes, and so on. The different spectral niches are exploited by different photosynthetic pigments of phytoplankton[11,15].

In the crystal lattice of ice, however, $H_2O$ molecules are more strongly hydrogen-bonded than in liquid water, resulting in a broadening and weakening of the overtone bands[29,30]. Hence, in ice, the peaks of the 8th, 7th, 6th, 5th and 4.1 vibrational harmonics are flattened out, resulting in a much smoother absorption spectrum across the visible range (Fig. 1b). Moreover, the peak due to the 4th harmonics is smaller and shifted to longer wavelengths in ice as compared to liquid water[20].

In view of these differences in scattering and absorption properties of ice and liquid water, it is likely that the disappearance of sea ice will change not only the intensity but also the spectral quality of light. Yet, exactly how light spectra for aquatic photosynthesis are affected by the loss of ice cover is still unclear.

In this study, we implement the optical properties of sea ice[17,20,31] and liquid water[23,25,32] into a state-of-the-art radiative transfer model[33],

to investigate how irradiance spectra below sea ice differ from irradiance spectra in open water (Fig. 1c, d). We focus our analysis on the euphotic zone (defined as the water layer extending from the surface to the depth at which light has been attenuated to 1% of the surface irradiance), where most photosynthetic activity takes place. Our results reveal that the loss of sea ice will cause marked changes in the spectral composition of light, which may alter the photosynthetic pigments favored by natural selection.

## Results

### General outline and model validation

We consider bare sea ice, representative of late spring and summer when the snow has melted. After snow melt, often a porous granular structure develops in the top layer of melting sea ice known as the surface scattering layer[19]. The transition zone below the surface scattering layer will be described as the drained layer, and the underlying columnar ice as the interior layer[19,31]. Our aim is to model the transmission of irradiance through sea ice to obtain the irradiance spectrum at the bottom of the ice, where dense assemblages of ice algae and under-ice phytoplankton blooms may develop (Fig. 1c). Irradiance spectra in sea ice and water were modeled with the radiative transfer model Hydrolight-Ecolight, version 6.0, which solves the radiative transfer equation numerically[32,33]. In our application, the model combines known optical properties of seawater, sea ice, chlorophyll $a$, colored dissolved organic matter (CDOM) and non-algal particles (NAP) with measurements of the incident solar irradiance spectrum, albedo, ice thickness and concentrations of chlorophyll $a$, CDOM and NAP (Supplementary Fig. S1 and Supplementary Table S1).

Although Hydrolight has been extensively validated for marine waters[32], less data are available on the performance of Hydrolight in

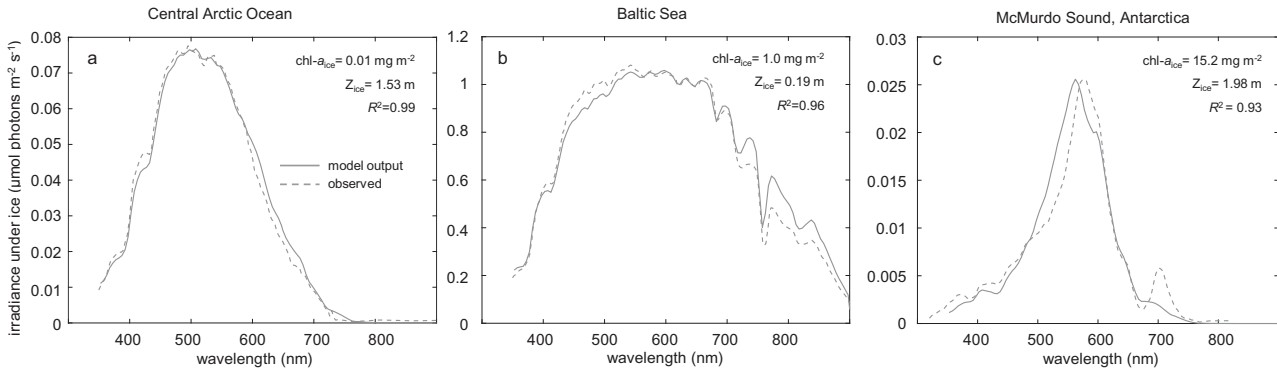

**Fig. 2 | Predicted and observed planar irradiance spectra under sea ice at three different locations. a** Central Arctic Ocean, **b** Baltic Sea, and **c** McMurdo Sound, Antarctica. Observed spectra (dashed lines) in (**a**) and (**c**) are from Lund-Hansen et al.[2] and in (**b**) from Kari et al.[34]. In each panel, the measured chlorophyll-*a* concentration in the ice (chl-$a_{ice}$), ice thickness ($z_{ice}$), and coefficient of determination ($R^2$) between the predicted and observed spectrum are indicated. Predicted spectra (solid lines) are based on Hydrolight-Ecolight 6.0, using the input data in Supplementary Fig. S1 and Supplementary Table S1.

areas covered with sea ice. To validate the model predictions, we therefore first compared light spectra under sea ice predicted by the radiative transfer model with measured spectra[2,34] under sea ice without snow cover at three different locations across the globe (Fig. 2).

In sea ice of the Central Arctic Ocean[2,35], light absorption by CDOM and NAP was minor ($a_{CDOM}(440) = 0.10\,m^{-1}$, $a_{NAP}(440) = 0.01\,m^{-1}$), and the chlorophyll *a* concentration of ice algae was very low (~0.01 mg m⁻²). The light spectrum underneath an ice cover of ~1.5 m spanned a broad range of wavelengths, from violet to orange light (Fig. 2a).

In the Baltic Sea[34], the sea ice was much thinner (19 cm), and hence irradiance levels under the ice cover of the Baltic Sea were much higher than in the Arctic Ocean. However, sea ice of the Baltic Sea contained more CDOM and NAP ($a_{CDOM}(440) = 0.39\,m^{-1}$, $a_{NAP}(440) = 0.05\,m^{-1}$) and a higher chlorophyll *a* concentration (1.0 mg m⁻²) than the Arctic Ocean. This implies that UV and violet light were more strongly absorbed than other wavelengths, and the irradiance spectrum under sea ice of the Baltic Sea was shifted to longer wavelengths (Fig. 2b).

At McMurdo Sound, in Antarctica[2], the ice was ~2 m thick, light absorption by both CDOM and NAP was low ($a_{CDOM}(440) = 0.06\,m^{-1}$; $a_{NAP}(440) = 0.01\,m^{-1}$), whereas the chlorophyll *a* concentration of ice algae was very high (15.2 mg m⁻²). Hence, both violet-blue and red wavelengths were largely absorbed by chlorophyll, and the light spectrum underneath the ice cover was dominated by green and yellow wavelengths (Fig. 2c).

In general, the modeled spectra show good agreement with the measured spectra ($R^2$ ranged from 0.93 to 0.99), confirming the robustness of the model predictions. Probably, the model predictions could be even further improved by fine-tuning the parameter values (e.g., by small changes in CDOM or chlorophyll concentration), but we preferred to base our model predictions as much as possible on measured parameter values (Supplementary Table S1).

### Comparison of light spectra in ice-covered versus open water
Now that the model has been successfully validated, we use the radiative transfer model to compare irradiance spectra in ice-covered water and in open water without sea ice (Figs. 3 and 4). In our first example, we assume an ice thickness of 1 m, with a surface scattering layer[19,31] but without snow cover. The model parameters for ice-covered and open water are otherwise identical, with the same incident solar irradiance (Supplementary Fig. S2) and very low CDOM and NAP concentrations representative of the clear oligotrophic ocean ($a_{CDOM}(440) = 0.01\,m^{-1}$; $a_{NAP}(440) = 0.001\,m^{-1}$). Concentrations of ice algae and phytoplankton are set to zero to predict irradiance spectra generated by the abiotic environment (i.e., by sea ice and seawater). In other words, we calculate spectra available for aquatic photosynthesis when new populations of ice algae or phytoplankton start to grow.

The model predicts that, in the first few centimetres below the surface, scalar irradiance (i.e., irradiance from all directions) is much higher in sea ice than in open water due to intense backscattering by ice (Fig. 3a, b). However, light penetrates less deep in sea ice than in open water. For example, integrated over the spectrum, only 20% of the total surface irradiance penetrates to a depth of 0.97 m in sea ice, whereas the same irradiance reaches 18.4 m depth in open ocean water (Fig. 3c, d). The more rapid light attenuation by sea ice is due to its high albedo and strong scattering in comparison to liquid water (Fig. 1; see also Supplementary Table S3). The model predictions also suggest differences in spectral composition of ice-covered versus open ocean water, although this is difficult to assess from Fig. 3a, b in view of the concomitant differences in total irradiance.

Most aquatic photosynthesis takes place in the euphotic zone, where light levels are sufficient to support primary production. We therefore investigated differences in spectral composition of the euphotic zone by comparing spectra of ice-covered versus open water at the same optical depth, i.e., at the same total irradiance integrated over the spectrum (350–900 nm). In this way, the shape of the spectra can be compared without potentially confounding effects of differences in total irradiance. The spectra are presented for five optical depths, at which the total scalar irradiance is attenuated to 20%, 10%, 5%, 2%, and 1% of the surface irradiance. The results reveal marked differences in spectral shape between ice-covered and open water (Fig. 3c, d). Light transmitted through ~1 m of sea ice spans essentially the entire photosynthetically active range from 400 to 700 nm (see the 20% optical depth in Fig. 3c). At 10% and 5% optical depth, water below sea ice still has a relatively broad irradiance spectrum that covers violet, blue and green wavelengths (400–600 nm; Fig. 3c), whereas open ocean water at the same optical depth is characterized by a more pronounced peak largely confined to the violet and blue part of the spectrum (400–500 nm; Fig. 3d). Further down in the euphotic zone, at 2% and 1% optical depth, the spectra in ice-covered and open ocean water become more similar, as the spectrum in the water column below sea ice gradually narrows down to violet-blue wavelengths as well. These results imply that, at least in this example, ice algae attached to the bottom of the ice and phytoplankton communities in the first several meters of water underneath the ice will be exposed to a broader range of wavelengths than phytoplankton communities at the same optical depths in open ocean water.

### Ecosystems with different CDOM concentrations
CDOM absorbs strongly in the violet and blue part of the spectrum, and is a major determinant of the light conditions in aquatic ecosystems[11,32]. We therefore investigated the spectral composition of light for three different marine ecosystems that span a wide range of

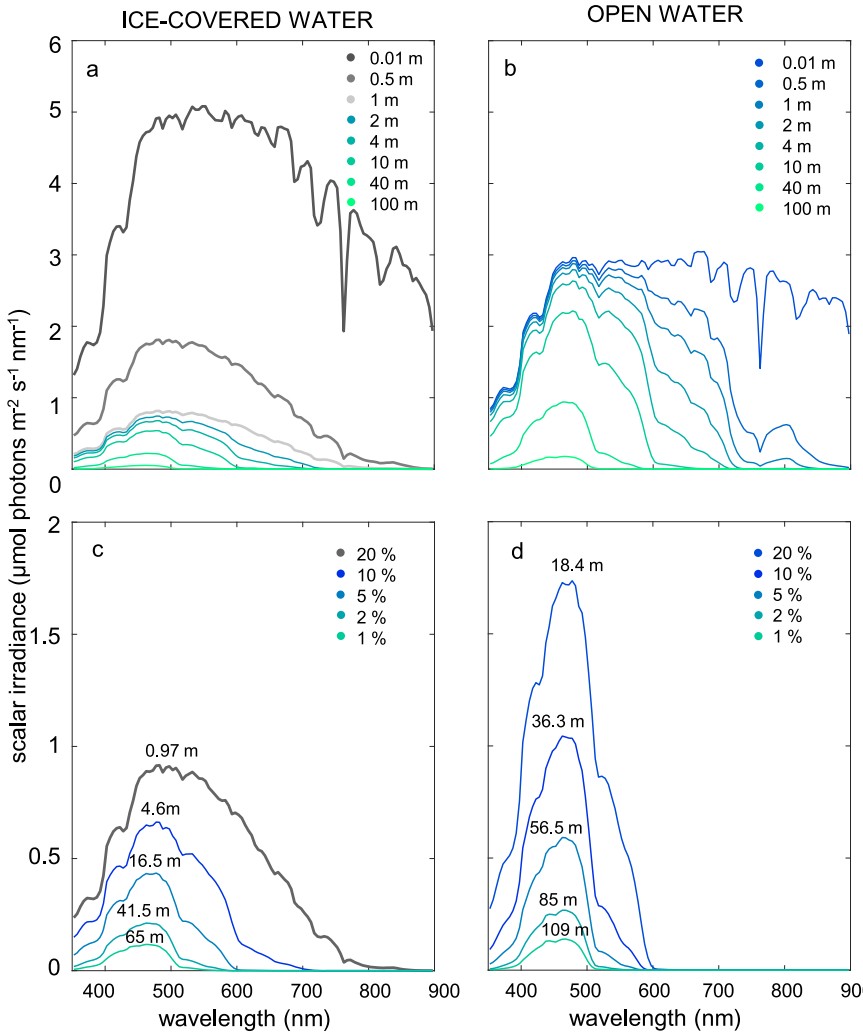

**Fig. 3 | Irradiance spectra in ice-covered versus open ocean water.** The graphs show scalar irradiance spectra predicted by the model for (**a**, **c**) ice-covered water with an ice thickness of 1 m, and (**b**, **d**) open water without ice cover. In (**a**, **b**), irradiance spectra are compared at the same physical depths. In (**c**, **d**), irradiance spectra are compared at the same optical depths (20%, 10%, 5%, 2%, and 1% of the incident solar irradiance). Gray lines indicate spectra in sea ice, while blue lines indicate spectra in water below the sea ice and in open water. All four panels assume the same incident solar irradiance spectrum (Supplementary Fig. S2) and the same parameter values (Supplementary Table S2), with very low concentrations of colored dissolved organic matter (CDOM) and non-algal particles (NAP) representative of the oligotrophic ocean (light absorption by CDOM and NAP at 440 nm is $a_{CDOM}(440) = 0.01\ m^{-1}$; $a_{NAP}(440) = 0.001\ m^{-1}$). The only difference between the panels is the ice cover.

CDOM concentrations, from oligotrophic ocean waters with very low CDOM concentrations to coastal waters with high CDOM concentrations (Fig. 4). For each of these marine ecosystems, we calculated irradiance spectra directly below sea ice and in open water, at several optical depths within the euphotic zone. Without sea ice, the clear oligotrophic ocean is dominated by violet-blue colors, whereas the underwater spectrum is shifted to green and yellow wavelengths in coastal waters with high CDOM concentrations (Fig. 4d–f).

For all three ecosystems, the results show that irradiance spectra directly below sea ice span a wider range of wavelengths than irradiance spectra at the same optical depth in open water without sea ice (Fig. 4). For example, at an optical depth of 25% of the surface irradiance, the full width at half maximum (FWHM) of the irradiance spectrum below an ice layer in the oligotrophic ocean is 280 nm, whereas the FWHM in open ocean water is only 140 nm (Fig. 4a, d and Table 1). Similarly, in coastal water, the FWHM is 285 nm below ice, but only 190 nm in open water without sea ice (Fig. 4c, f and Table 1). Hence, the loss of sea ice generally results in a narrowing of the irradiance spectra in the euphotic zone, irrespective of the CDOM concentration in the ecosystem.

Furthermore, at an optical depth of 25%, the mean wavelength of the irradiance spectrum in the oligotrophic ocean is shifted from the green part of the spectrum (550 nm) below sea ice to the blue part of the spectrum (472 nm) in open ocean water (Fig. 4a, d, and Table 1). This is a blue shift of almost 80 nm. Conversely, in coastal water, the mean wavelength is shifted from 605 nm below sea ice to 582 nm in open water, which is a shift of less than 25 nm (Fig. 4c, f and Table 1). Hence, the blue shift to shorter wavelengths due to the loss of sea ice is more pronounced in oligotrophic ocean waters with low CDOM concentrations than in coastal waters with high CDOM concentrations.

### Variation in optical properties of sea ice

The albedo and scattering coefficient of sea ice vary widely among different ice types, depending on the presence of a surface scattering layer, melt ponds, the number and size distribution of air bubbles and brine inclusions, as well as ice temperature[20,31,36]. To investigate the sensitivity of our model predictions, we therefore compared irradiance spectra below sea ice predicted for different surface albedos (Supplementary Fig. S3) and different scattering coefficients (Supplementary Fig. S4). The results show that sea ice with a high albedo and

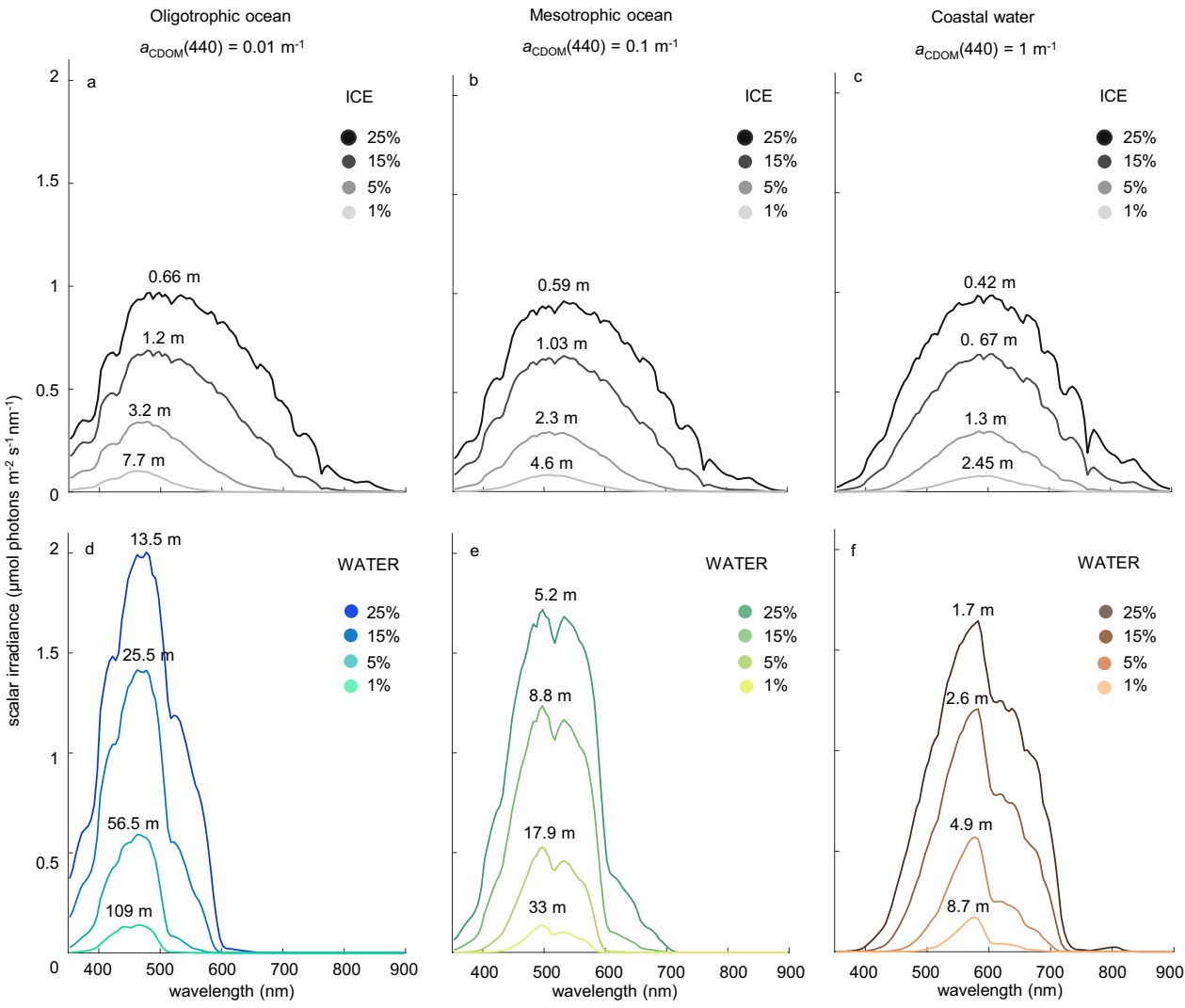

**Fig. 4 | Irradiance spectra below sea ice and in open water, predicted for three marine ecosystems.** The graphs show model predictions of scalar irradiance spectra (**a**–**c**) directly under the ice (gray lines) and (**d**–**f**) in open water (colored lines). The results are shown for three concentrations of colored dissolved organic matter (CDOM), representative for the oligotrophic ocean (left column), mesotrophic ocean (middle column), and eutrophic coastal waters (right column) ($a_{CDOM}$(440) is light absorption by CDOM at 440 nm). In each panel, irradiance spectra are shown at the same four optical depths (25%, 15%, 5%, and 1% of the incident irradiance). The associated physical depths (in meters) are also indicated, where the physical depth represents the thickness of the ice in (**a**–**c**) and the depth in the water column in (**d**–**f**). The incident solar irradiance spectrum is the same in all panels (see Supplementary Fig. S2), and parameter settings are summarized in Supplementary Table S2.

high scattering coefficients will generate slightly broader irradiance spectra than sea ice with a low albedo and low scattering coefficients.

Furthermore, in early spring, sea ice is often covered by a layer of snow[6,19]. Snow consists of ice crystals and air. Repeated scattering at numerous air-ice interfaces gives snow a very high albedo, and its wavelength-dependent light attenuation can strongly diminish light levels[37–39]. Our calculations revealed that even a thin layer of only 3 cm of snow can remove almost 90% of the incident irradiance, in line with field observations[21,39]. Yet, low light levels below snow-covered ice can already provide sufficient irradiance for ice algae to grow[40,41]. We therefore also compared irradiance spectra below ice predicted for different degrees of snow cover, and extended the comparison to lower light levels of only 0.1% of the surface irradiance. The results show that sea ice covered by snow will generate even broader irradiance spectra than bare sea ice without snow cover (Supplementary Fig. S5).

Overall, however, the patterns predicted by the model are robust. That is, irradiance spectra in the euphotic zone are narrowed down to shorter wavelengths in open water than below sea ice, across the full range of albedos, scattering coefficients, and snow cover investigated (Supplementary Figs. S3–5).

## Spectral niches for aquatic photosynthesis

Next, we expand our comparison between the spectral composition of sea ice and water from only three marine ecosystems (Fig. 4) to a large number of marine ecosystems (Fig. 5). First, we calculated 85 total absorption spectra, as the sum of the absorption spectra of either sea ice or pure seawater, with a small amount of NAP, and 85 different CDOM concentrations. These CDOM concentrations cover the full range from the clearest ocean waters with almost no CDOM to humic-rich plumes of Arctic rivers with very high CDOM concentrations obtained from the vast amounts of organic carbon stored in the Arctic

**Table 1 | Properties of the spectra in Fig. 4 at optical depths of 25% and 1% of the surface irradiance, including their full width at half maximum (FWHM), peak wavelength, and mean wavelength**

| Ecosystem | | SEA ICE | | | OPEN WATER | | |
|---|---|---|---|---|---|---|---|
| | $a_{CDOM}(440)^a$ (m$^{-1}$) | FWHM (nm) | Peak wavelength (nm) | Mean wavelength (nm) | FWHM (nm) | Peak wavelength (nm) | Mean wavelength (nm) |
| 25% optical depth | | | | | | | |
| Oligotrophic ocean | 0.01 | 280 | 498 | 550 | 140 | 478 | 472 |
| Mesotrophic ocean | 0.1 | 290 | 533 | 562 | 150 | 498 | 514 |
| Coastal water | 1 | 285 | 608 | 605 | 190 | 583 | 582 |
| 1% optical depth (euphotic depth) | | | | | | | |
| Oligotrophic ocean | 0.01 | 105 | 463 | 464 | 80 | 468 | 452 |
| Mesotrophic ocean | 0.1 | 120 | 508 | 515 | 85 | 498 | 511 |
| Coastal water | 1 | 125 | 603 | 592 | 50 | 578 | 576 |

$^a a_{CDOM}(440)$ is light absorption by colored dissolved organic matter (CDOM) at 440 nm.

tundra[42,43] (Fig. 5a, b). Similarly, we also calculated total scattering spectra for sea ice and liquid water (note that light is not scattered by CDOM). Subsequently, these absorption and scattering spectra were implemented in the radiative transfer model to predict the corresponding irradiance spectra at the euphotic depth (i.e., the depth at which irradiance over the entire spectrum equals 1% of surface irradiance) (Fig. 5c, d).

An overlay of all 85 irradiance spectra below sea ice reveals a nearly flat landscape without major valleys from 450 to 750 nm (Fig. 5c). This flat landscape is due to the smooth absorption spectrum of ice across the visible range and into the near-infrared (Fig. 5a). Hence, sea ice offers a continuum of irradiance spectra, in which the harmonics of the vibrational modes of H$_2$O do not play a role.

In contrast, overlaying the 85 irradiance spectra of open water without ice cover reveals a rugged landscape of peaks and valleys (Fig. 5d). The deep valleys are induced by the harmonics of the vibrational modes of liquid water[11,15] (Fig. 5b). They delineate a series of distinct spectral niches. Clear ocean waters have very low CDOM concentrations, and their underwater spectra show a major peak in the violet niche between the 8th and 7th harmonics (401–449 nm) and another major peak in the blue niche between the 7th and 6th harmonics (449–514 nm) (Fig. 5d). Coastal waters with intermediate to high CDOM concentrations have underwater spectra with a major peak in the green niche between the 6th and 5th harmonics (514–605 nm). River plumes with very high CDOM concentrations have underwater spectra in the orange and red niche between the 5th harmonics and 4.1 sub-harmonic (605–662 nm) and the 4.1 sub-harmonic and 4th harmonics (662–742 nm), respectively.

## Discussion

Our analysis reveals three major results. First, the transition from ice cover to open water leads to narrower irradiance spectra in the euphotic zone (Figs. 3, 4 and Table 1). This difference in spectral width can be attributed to the high albedo of sea ice and to differences in the relative importance of absorption versus scattering (Fig. 1a, b and Supplementary Table S3). The high albedo of sea ice removes a large fraction of the incident solar radiation in a wavelength-independent manner. Furthermore, throughout the photosynthetically active range (400–700 nm), light attenuation in sea ice is dominated by scattering due to air bubbles and brine inclusions, which is also essentially wavelength independent. Conversely, light attenuation in liquid water is largely dominated by absorption, which varies markedly with wavelength and thereby selectively removes part of the incident solar spectrum. Hence, at the same optical depth, irradiance spectra below an ice layer span a relatively wide range of wavelengths broadly resembling the incident solar spectrum, whereas irradiance spectra in liquid water have been narrowed by absorption.

Second, the transition from sea ice to open water shifts the irradiance spectrum in the euphotic zone to shorter wavelengths. Light attenuation by sea ice is dominated by scattering, which is largely wavelength-independent. In contrast, light absorption by open ocean water selectively removes the longer wavelengths (orange, red, and infrared) while the shorter wavelengths (violet and blue) remain. This blue shift is especially pronounced in ocean ecosystems with a low CDOM concentration (Fig. 4a, d). Since CDOM absorbs relatively strongly at short wavelengths, the blue shift due to the loss of ice cover is less pronounced or absent in marine ecosystems with high CDOM concentrations (e.g., coastal waters, river plumes).

Third, our analysis shows that ice-covered regions offer a continuum of irradiance spectra, whereas open water offers a series of distinct spectral niches (Fig. 5). This is caused by differences in the smoothness of the absorption spectra of ice versus liquid water. Due to the smooth absorption spectrum of ice, variation in the CDOM concentration generates a continuum of irradiance spectra across the entire photosynthetically active range. In contrast, local peaks in the absorption spectrum of liquid water selectively remove specific wavelengths, and thereby generate several well-defined spectral niches in the lakes and oceans of our planet that are separated by the vibrational harmonics of the water molecule[11,15,28]. Hence, the transition from sea ice to open water turns a continuum of irradiance spectra into either a violet, blue, green, orange, or red spectral niche depending on the CDOM concentration.

What are the implications for aquatic photosynthesis? Based on our findings, one should expect that phytoplankton of the open ocean will be specialized on a relatively narrow range of short wavelengths, while ice algae will thrive at a broader range of wavelengths. This expectation may help to explain why ice algae, and also phytoplankton blooms under the ice, are often dominated by diatoms[5,44,45]. Diatoms utilize chlorophyll *a*, chlorophyll *c* and the accessory pigment fucoxanthin for photosynthesis, which provides them with a smooth absorption spectrum (without indents from the vibrational harmonics of liquid water) spanning a broad range of violet, blue and green wavelengths from 400 to 550 nm[44,46] (Fig. 6a). Prymnesiophytes of the *Phaeocystis* genus are also common in and below sea ice, both in the Arctic Ocean (*P. pouchetii*) and in Antarctic waters (*P. antarctica*)[47,48]. They also combine chlorophylls *a* and *c* and several fucoxanthins[49,50], which provides them with a similarly broad absorption spectrum as the diatoms.

In contrast, many (but not all) phytoplankton taxa of the open ocean have absorption spectra with relatively slender peaks targeting specific wavebands (Fig. 6b). For example, the small prasinophyte *Micromonas polaris* is highly abundant in open waters of the oligotrophic Arctic Ocean[51,52] and uses chlorophyll *a* and *b*, which capture the violet niche (between the 8th and 7th harmonic) and blue niche

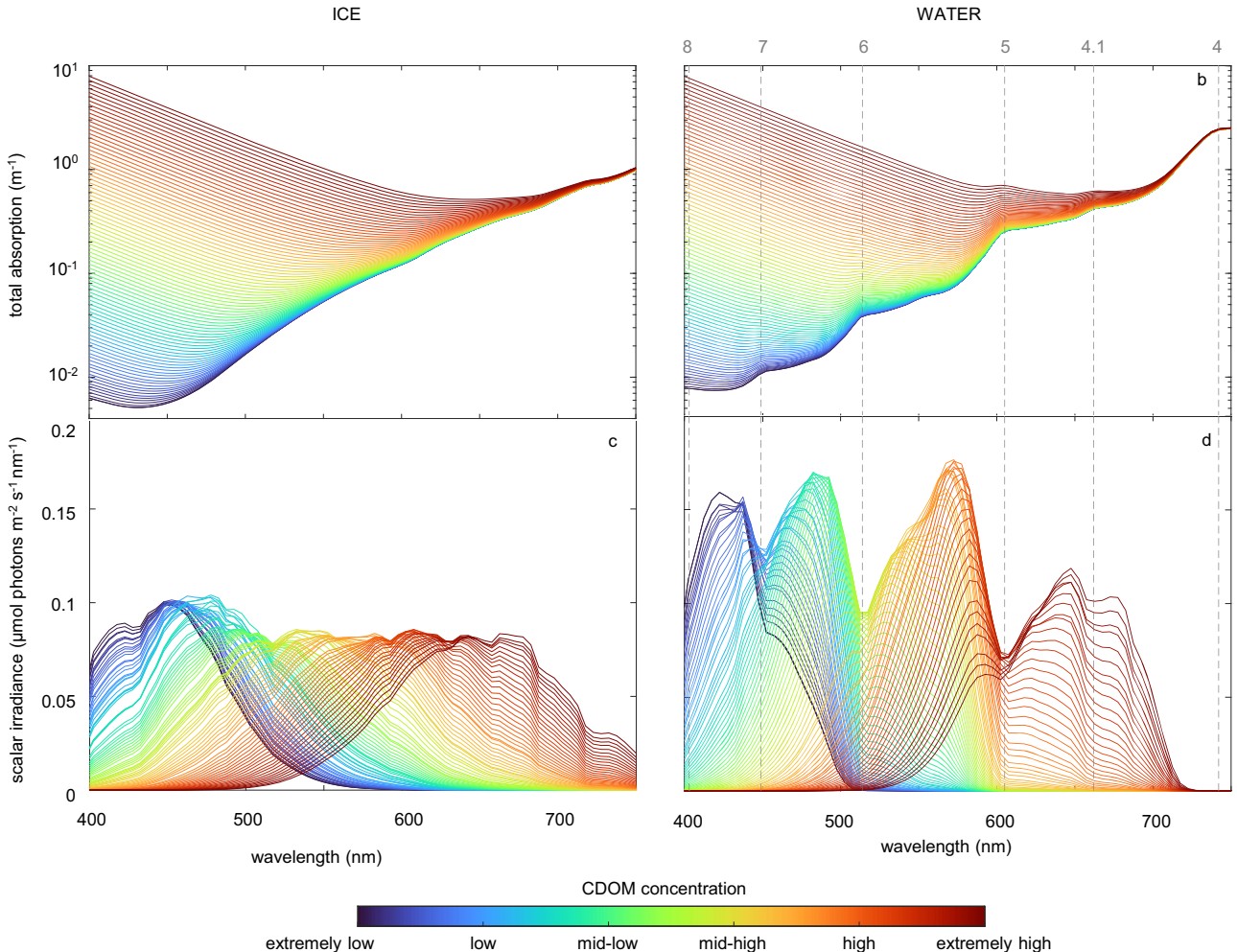

**Fig. 5 | 'Spectral landscapes' below sea ice and in open water predicted by the model. a, b** Total absorption spectra of sea ice (**a**) and open water (**b**) at different concentrations of colored dissolved organic matter (CDOM). **c, d** Irradiance spectra directly below sea ice (**c**) and in open water (**d**), at the euphotic depth (1% light level), obtained from the corresponding total absorption spectra. Each panel shows 85 absorption or irradiance spectra, spanning a wide range of CDOM concentrations from the clearest ocean waters ($a_{CDOM}(440) = 0.0025\ m^{-1}$) to the humic-rich river plumes of large Arctic rivers ($a_{CDOM}(440) = 5\ m^{-1}$). Vertical dashed lines in (**b**, **d**) indicate the harmonics of the vibrational modes of $H_2O$. The incident solar irradiance spectrum is the same for all graphs (see Supplementary Fig. S2), and parameter settings are summarized in Supplementary Table S2.

(between the 7th and 6th harmonic), respectively[15]. The absorption peaks of these two chlorophylls are clearly separated by the 7th harmonic at 449 nm (Fig. 6b). Moreover, prasinophytes lack fucoxanthins. Hence, contrary to diatoms and *Phaeocystis* spp., prasinophytes do not extend their absorption far into the green part of the spectrum. Cryptophytes are also common in polar seas and can be found in both sea ice and open water[45]. They combine chlorophyll *a* and *c* with the phycobiliprotein phycoerythrin (PE) or phycocyanin (PC) and several chromophores. The pigmentation of cryptophytes is classified into eight phycobiliprotein classes, with absorption peaks at different wavelengths in the range from 545 to 645 nm[53]. Some examples are shown in Fig. 6b. Phycobiliprotein classes Cr-PE 545, Cr-PE 555, and Cr-PC 577 have their absorption peak in the green niche between the 6th and 5th harmonic, whereas Cr-PC 645 has its absorption peak in the orange niche between the 5th harmonic and 4.1 subharmonic (605–662 nm). Finally, we note that the red peak of chlorophyll *a*, which is shared by all oxygenic photosynthetic organisms, captures the red niche between the 4.1 subharmonic and 4th harmonic (662–742 nm). Hence, prasinophytes, cryptophytes, and several other phototrophic organisms of the open ocean (e.g., cyanobacteria) have tuned the peaks in their absorption spectra to the spectral niches delineated by the vibrational harmonics of liquid water[11,15].

Accordingly, our model predicts that the loss of sea ice will induce changes in the pigment and species composition of phytoplankton communities. For example, in ice-covered waters of the oligotrophic ocean at, say, 5% of the incident light, the predicted spectrum spans a wide range of wavelengths from 400 to almost 600 nm (Fig. 4a). This wide range would be highly suitable for the photosynthetic pigments of diatoms and *Phaeocystis* spp. absorbing across the violet, blue, and green parts of the spectrum. In ice-free waters of the oligotrophic ocean, however, the corresponding predicted spectrum is narrowed to shorter wavelengths of 420–500 nm (Fig. 4d), which would be in favor of the prasinophyte *Micromonas* absorbing the violet and blue colors of the open ocean. These predictions are consistent with observed large-scale shifts from diatoms to *Micromonas* spp. in the oligotrophic Canada Basin of the Arctic Ocean[52,54], where ice cover has diminished considerably in recent decades[55].

In coastal waters of the West Antarctic Peninsula (WAP), cryptophytes are gradually increasing at the cost of other phytoplankton groups[56,57]. Our model predicts that, in coastal waters with a high CDOM concentration, ice cover will yield a broad spectrum ranging from 500 to 700 nm (Fig. 4c, again at 5%), spanning the absorption peaks of all eight phycobiliprotein classes of cryptophytes. In contrast, the predicted spectrum of ice-free coastal waters is much narrower. At

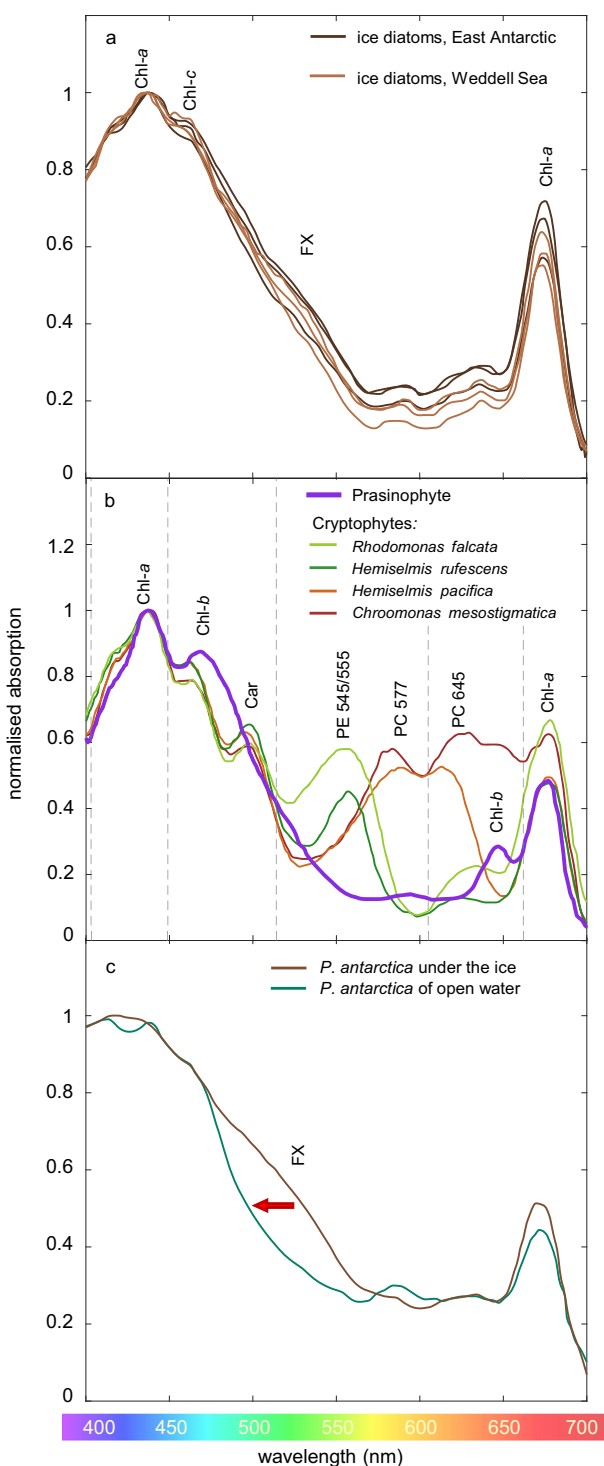

**Fig. 6 | Absorption spectra of ice algae and phytoplankton. a** Natural assemblages of ice diatoms collected from the East Antarctic (dark brown lines) and the Weddell Sea (light brown lines). Spectral data are from Dalman et al.[44]. **b** Cultures of a prasinophyte (here we show *Ostreococcus tauri*[81], which has a similar pigmentation as *Micromonas* spp.) and the cryptophytes *Rhodomonas falcata* NIES702 (phycobiliprotein class Cr-PE 545)[53], *Hemiselmis rufescens* CCMP440 (Cr-PE 555)[53], *Hemiselmis pacifica* CCMP706 (Cr-PC 577)[53], and *Chroomonas mesostigmatica* CCMP1168 (Cr-PC 645)[82]. Vertical dashed lines indicate the 8th, 7th, 6th, and 5th harmonics (at 401, 449, 514, and 605 nm) and the 4.1 sub-harmonic (at 662 nm) of the vibrational modes of $H_2O$. **c** *Phaeocystis antarctica* collected from under sea ice (brown line) and from open water (green line) at McMurdo Sound, Antarctica. Spectral data are from Beeler SooHoo et al.[66]. All absorption spectra were normalized to their maximum value, after baseline correction for minimum absorbance at 750 nm. Abbreviations: Chl chlorophyll, FX fucoxanthin, Car carotenoid, PE phycoerythrin, PC phycocyanin.

Of course, many other factors affect the species composition of phytoplankton communities in polar seas as well. For example, Ardyna et al.[59] reported that under-ice phytoplankton blooms of the Arctic Ocean are dominated by two different phytoplankton assemblages (diatoms or *Phaeocystis*) depending on the nitrate:silicate ratio, which may result in an increase of *Phaeocystis pouchetii* and a decrease of diatom-dominated blooms as silicate inputs from the subpolar Atlantic Ocean decline[6,60,61]. In open waters of the Canada Basin, melting sea ice and warming of surface waters have resulted in a strong density stratification, which limits the supply of nitrogen and other nutrients from the deep into the surface layer[52,62]. These nutrient-poor conditions favor a shift from diatoms to picoeukaryotes such as *Micromonas* spp, as their very small size provides a competitive advantage when nutrients are limiting and diminishes their sinking rate[52,63]. The shift from diatoms to cryptophytes in coastal waters of the WAP has been attributed to a shallowing of the surface mixed layer[57], a phenomenon again associated with enhanced stratification due to ice melt. Furthermore, several cryptophytes do not rely exclusively on autotrophic growth enabled by their photosynthetic pigments, but can also obtain carbon and nutrients from mixotrophic feeding on bacterial populations in CDOM-rich waters[64,65]. Hence, it would be naive to presume that the light spectrum is the only selective factor in polar waters, or that there will be a perfect match between irradiance spectra and the pigment composition of phototrophic communities. Nevertheless, the overall correspondence between changes in irradiance spectra due to the loss of sea ice and changes in pigmentation of ice algae versus open ocean phytoplankton is remarkable, indicating that light quality is an important selective factor in communities of photosynthetic organisms.

The loss of ice cover may also induce adaptive changes in pigment composition within species. Beeler Soohoo et al.[66] compared in vivo absorption spectra of planktonic *Phaeocystis antarctica* (at that time still called *P. pouchetii*) collected under sea ice and from open water at the ice edge of McMurdo Sound, Antarctica. Irradiance levels below sea ice were low, and the spectrum was dominated by green wavelengths due to light attenuation by sea ice and self-shading by high densities of ice algae. Their study did not measure the irradiance spectrum in open water, but most likely it was shifted to shorter wavelengths (as in Fig. 4d, e). The absorption spectrum of *P. antarctica* under the ice spanned the violet and blue wavelengths, and also showed a pronounced shoulder from 490 to 550 nm, which signifies light absorption by fucoxanthins in the green part of the spectrum (Fig. 6c). In contrast, *P. antarctica* sampled from the open water mainly absorbed violet and blue wavelengths (<514 nm), but lacked this green shoulder, indicating that it had a much lower fucoxanthin content. Thus, in line with our model predictions, *P. antarctica* of open water showed a narrower absorption spectrum limited to shorter wavelengths than *P. antarctica* under sea ice (Fig. 6c). Yet, interpretation of these findings is not straightforward. Algae exposed to high

the CDOM concentration used in our example, the slender irradiance peak is positioned at 578 nm (Fig. 4f), which is mainly suitable for cryptophytes of type Cr-PC 577 (Fig. 6b). At lower CDOM concentrations, this slender peak will shift to shorter wavelengths, which will favor another cryptophyte pigment type (e.g., Cr-PE 545). Hence, these model results suggest that the loss of ice cover will cause a decrease in the diversity of cryptophyte pigment types. This prediction is in agreement with the low cryptophyte diversity observed in open waters along the WAP[58]. It might therefore be interesting for future studies to look more closely at the diversity and biogeography of the different pigment types of cryptophytes in ice-covered versus open waters of coastal regions.

irradiances near the water surface tend to have low contents of accessory photosynthetic pigments (including fucoxanthin) and high contents of photoprotective pigments to avoid photodamage[7,8,67]. Enhanced fucoxanthin contents are therefore often interpreted as a photoacclimative response to low irradiances, which may explain the high fucoxanthin contents under ice reported by Beeler Soohoo et al.[66]. At these low irradiances, however, acclimation to the prevailing light spectrum below sea ice likely also plays a role, as enhanced fucoxanthin contents clearly extend light harvesting for photosynthesis to longer wavelengths.

In conclusion, the transition from ice-covered to open water alters the colors of light available for aquatic photosynthesis by narrowing down irradiance spectra to shorter wavelengths and generating more pronounced spectral niches. These results imply that the loss of sea ice by global warming will likely lead to major changes in the pigment composition, and thereby also in the taxonomic composition, of primary producers in polar marine ecosystems.

## Methods

### Radiative transfer model

Irradiance spectra in ice-covered water (Fig. 1c) and open water (Fig. 1d) were modeled with Hydrolight-Ecolight version HE6.0, a state-of-the-art radiative transfer model that solves the radiative transfer equation numerically[32,33]. The radiative transfer equation describes how the radiance changes with depth in the sea ice or water column, depending on the inherent optical properties (IOP) of different components[32,68]. Depth-dependent scattering by sea ice is incorporated by assuming three distinct layers in the sea ice model[19,69]: a surface scattering layer (SSL) with high scattering, a drained layer (DL) with intermediate scattering, and interior ice (INT) with relatively low scattering (Supplementary Fig. S6).

For a horizontally homogeneous water body or ice layer with incident light from above, and negligible fluorescence and bottom reflectance, the change in spectral radiance $L$ (W m$^{-2}$ sr$^{-1}$ nm$^{-1}$) with depth $z$ at wavelength $\lambda$ is described by:

$$\mu \frac{dL(\lambda, z, \theta, \varphi)}{dz} = -c(\lambda, z)L(\lambda, z, \theta, \varphi) + g(\lambda, z, \theta, \varphi) \quad (1)$$

where $\theta$ and $\varphi$ are the zenith angle and azimuth angle, respectively, and $\mu = \cos(\theta)$ is the vertical projection of radiance $L$. The term $c(\lambda, z)$ is the beam attenuation coefficient (m$^{-1}$) describing the radiance lost due to absorption and scattering when passing through the water column or ice layer, and $g(\lambda, z, \theta, \varphi)$ is the radiance gained with depth due to scattering from adjacent paths.

The beam attenuation coefficient is defined as the sum of the total absorption coefficient $a(\lambda, z)$ (m$^{-1}$) and total scattering coefficient $b(\lambda, z)$ (m$^{-1}$):

$$c(\lambda, z) = a(\lambda, z) + b(\lambda, z) \quad (2)$$

We defined five optically significant components in the model: seawater (sw), sea ice (ice), CDOM, NAP, and photosynthetic organisms, including ice algae and phytoplankton (algae). The total absorption and scattering coefficient are calculated from the absorption and scattering properties of these optical components:

$$a(\lambda, z) = a_{sw}(\lambda, z) + a_{ice}(\lambda, z) + a_{CDOM}(\lambda, z) + a_{NAP}(\lambda, z) + a_{algae}(\lambda, z) \quad (3)$$

and

$$b(\lambda, z) = b_{sw}(\lambda, z) + b_{ice}(\lambda, z) + b_{NAP}(\lambda, z) + b_{algae}(\lambda, z) \quad (4)$$

The contribution of CDOM to scattering is assumed negligible[32].

The model calculates both planar and scalar irradiance spectra. Downwelling planar irradiance $E_d(\lambda, z)$ (in W m$^{-2}$ nm$^{-1}$) is often measured by field studies, including the studies we used for our model validation (see below). It is defined as the total radiance, at a given wavelength and depth, from the upper hemisphere with respect to solid angle $\omega$:

$$E_d(\lambda, z) = \int_{\omega=0}^{2\pi} L(\lambda, z, \omega) \cos(\theta) d\omega \quad (5)$$

For randomly oriented photosynthetic cells, however, photons from all directions can be used with equal efficiency. Scalar irradiance $E_o(\lambda, z)$ (W m$^{-2}$ nm$^{-1}$) is therefore a better all-round measure of the availability of light for photosynthesis[68]. It is defined as the total radiance, at a given wavelength and depth, from all directions:

$$E_o(\lambda, z) = \int_{\omega=0}^{4\pi} L(\lambda, z, \omega) d\omega \quad (6)$$

Since photosynthesis is based on the absorption of photons, the irradiance units were converted from energy flux (W m$^{-2}$ s$^{-1}$ nm$^{-1}$) into photon flux (photons m$^{-2}$ s$^{-1}$ nm$^{-1}$). For our model analysis, we calculated scalar irradiance spectra at four optical depths, receiving 25%, 15%, 5%, and 1% of the incident solar irradiance (expressed as photon flux, integrated over 350–900 nm), respectively. The 1% optical depth is also known as the euphotic depth.

### Inherent optical properties used in the model

*Seawater.* Absorption by pure water, $a_w(\lambda)$, is lowest at 344 nm and increases rapidly toward longer wavelengths, reaching the highest value in the near-infrared (NIR) range[22–25]. We used the absorption coefficients of Pope and Fry[23] for wavelengths above 550 nm and from ref. 25 for wavelengths shorter than 550 nm (Fig. 1b), assuming that salinity has a negligible effect on the absorption spectrum of water[25].

Elastic scattering by water molecules is inversely proportional to, roughly, the fourth power of the wavelength $\left(b_w \propto \lambda^{-4}\right)$ resulting in much stronger scattering in the blue than in the red and NIR range. Specifically, we computed scattering coefficients of pure water according to Morel[22]:

$$b_w(\lambda) = 16.06 \beta_w(90°; \lambda_0) \left(\frac{\lambda_0}{\lambda}\right)^{4.32} \quad (7)$$

with $\lambda_0 = 550$ and $\beta_w(90°; 550) = 0.93 \times 10^{-4}$. The scattering coefficient of pure water was adjusted for salinity $S$ to calculate the scattering coefficient of seawater[22,70]:

$$b_{sw}(\lambda) = \left(1 + 0.3\left(\frac{S}{37}\right)\right) b_w(\lambda) \quad (8)$$

where we assumed $S = 32$ g kg$^{-1}$ as default value and a low salinity of $S = 3.1$ g kg$^{-1}$ for the Bothnian Bay in the Baltic Sea[34].

*Sea ice.* Absorption coefficients of sea ice in the visible and NIR part of the spectrum were obtained from Warren & Brandt[71].

Scattering of light in sea ice is strongly affected by air bubbles, brine inclusions, and cracks in the ice. The number and size distribution of air bubbles and brine inclusions varies among different ice types, and the number of bubbles in sea ice often decreases with depth. We used a spectrally flat scattering coefficient of natural sea ice, with different values of $b_{ice}$ for different layers, ranging from $b_{ice} = 1$ m$^{-1}$ to $b_{ice} = 1000$ m$^{-1}$ depending on the depth and location (exact values in Supplementary Tables S1 and S2)[21,31,69]. To describe the angular distribution of the scattered light, we used the Petzold average-particle phase function, which corresponds to a backscattering ratio of 0.018

(ratio of backscattering to total scattering, $B = b_b/b$) and asymmetry parameter of 0.92[32,72].

CDOM. Absorption by CDOM decreases exponentially with increasing wavelengths:

$$a_{CDOM}(\lambda) = a_{CDOM}(\lambda_0)\exp^{-S_{CDOM}(\lambda-\lambda_0)} \qquad (9)$$

where $a_{CDOM}(\lambda_0)$ is the absorption by CDOM at reference wavelength $\lambda_0$. The exponential slope $S_{CDOM}$ varies slightly for different waters[73,74], with values ranging from 0.004 to 0.036 nm$^{-1}$. We used $\lambda_0 = 440$ nm as the reference wavelength and $S_{CDOM} = 0.014$ nm$^{-1}$ for the slope. The value of $a_{CDOM}(440)$ depends on the CDOM concentration. Our calculations were made for a wide range of CDOM concentrations, from $a_{CDOM}(440) = 0.0025$ m$^{-1}$ in very clear ocean waters with extremely low CDOM concentrations[75] to $a_{CDOM}(440) = 5$ m$^{-1}$ in the river plumes of Arctic rivers with very high CDOM concentrations[42,43].

NAP. NAP includes both organic particles (e.g., detritus) and inorganic particles (e.g., clay minerals) suspended in the water column or ice layer. Similar to CDOM, absorption by NAP decreases exponentially with wavelength:

$$a_{NAP}(\lambda) = a_{NAP}(\lambda_0)\exp^{-S_{NAP}(\lambda-\lambda_0)} \qquad (10)$$

where $\lambda_0$ is again the reference wavelength at 440 nm. The spectral slope $S_{NAP}$ ranges from 0.006 to 0.017 nm$^{-1}$, depending on the composition of the NAP[74,76]; we used $S_{NAP} = 0.007$ nm$^{-1}$. The value of $a_{NAP}(440)$ increases with the NAP concentration, as described by the empirical relation[72] $a_{NAP}(440) = 0.031$NAP; it ranges from 0.001 m$^{-1}$ in very clear ocean waters to sometimes more than 1 m$^{-1}$ in turbid coastal waters[74,76].

While CDOM has negligible scattering, NAP can be an important scattering component depending on the concentration, composition, and size of the suspended particles[77,78]. Here, we assume a mass-specific scattering function based on the average of the mass-specific scattering spectra of calcareous sand, red clay, yellow clay, and brown earth[33]. We used again the Petzold average-particle phase function to describe the angular distribution of the scattered light[32,72].

*Phytoplankton and ice algae*. Due to their photosynthetic pigments, phytoplankton and ice algae can strongly affect the absorption and scattering properties of aquatic systems. We included measured chlorophyll concentrations in the model for our model validation (Fig. 2), using the chlorophyll-specific spectral absorption and scattering properties described in the literature[79,80]. However, in the remainder of this study, we are interested in the light field created by the abiotic environment, which provides a template for the initial development of new populations of phytoplankton and ice algae. Therefore, concentrations of phytoplankton and ice algae, and hence their absorption and scattering, were set to zero in our model calculations for Figs. 3–5.

## Snow cover

For snow-covered ice in Supplementary Fig. S5, we assume a two-layer system with a snow layer of thickness $z_S$ on top of an interior layer of ice (INT)[19]. Planar irradiance $E_d(\lambda, z_S)$ transmitted through the snow layer was calculated by a simple semi-empirical model[37]:

$$E_d(\lambda, z_S) = (1-\alpha)E_d(\lambda, 0)e^{-\kappa(\lambda)z_S} \qquad (11)$$

where $\alpha$ is the albedo of snow, $E_d(\lambda, 0)$ is the incident solar irradiance, and $\kappa(\lambda)$ is the wavelength-dependent extinction coefficient of snow. We used the incident solar irradiance of Supplementary Fig. S2 and the albedo and wavelength-dependent extinction coefficient of snow measured by Perovich[37], taking into account the thickness of the snow layer. The planar irradiance transmitted through the snow served as input for the ice underneath.

## Model validation with field data

To validate the predictions of the radiative transfer model for ice-covered waters, we compare the model predictions with measured planar irradiance spectra below sea ice (Fig. 2). For this purpose, we used published field measurements from three locations with different optical properties: Amundsen Basin (station 231) in the Central Arctic Ocean[2,35], the Bothnian Bay (station D3) in the Baltic Sea[34] and McMurdo Sound in Antarctica (Figure 6.15 in ref. 2). At each location, planar irradiance spectra, $E_d(\lambda)$, in the visible and NIR range were measured both above the snow-free ice surface and below sea ice with hyperspectral TriOS RAMSES radiometers. Field measurements[2,34,35] of planar irradiance spectra above the sea ice (Supplementary Fig. S1), albedo of the snow-free ice surface, ice thickness, and concentrations of chlorophyll *a*, CDOM, and NAP served as input for the model.

NAP concentrations at the Central Arctic Ocean and McMurdo Sound were not reported in Lund-Hansen et al.[2,35]. Instead, we used $a_{NAP}(440) = 0.01$ m$^{-1}$ measured in Arctic sea ice by ref. 76. Furthermore, the CDOM concentration at McMurdo Sound was not measured by ref. 2, and scattering coefficients of sea ice were not measured at any of the three locations. Hence, $a_{CDOM}(440)$ at McMurdo Sound and the scattering coefficients of sea ice were estimated by least-squares fits of the model to the measured spectra. Estimates of the scattering coefficients at the Central Arctic Ocean and McMurdo Sound were constrained by the range of values reported for sea ice by ref. 31, while we allowed lower scattering coefficients for the Baltic Sea because of its low salinity[34].

For each location, the parameter values and functions used in the radiative transfer model are specified in Supplementary Table S1.

## Data availability

The spectral data generated in this study have been deposited in Figshare: https://figshare.com/s/abe54f20cd0165596170.

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

## Acknowledgements

We thank Elina Kari for sharing data collected in the Baltic Sea for Fig. 2b, and Maciej Soja for support with Matlab scripting. This research was funded by the ORIGINS project of the Faculty of Science of the University of Amsterdam.

## Author contributions

J.H. and S.W. conceived the original idea. J.H. designed the study in collaboration with M.S.W. The radiative transfer model was run by M.S.W., and the model results were analyzed by M.S.W. and J.H. Data for Fig. 2a, c, and Supplementary Fig. S1 were provided by L.C.L.H. The figures were made by M.S.W. The manuscript was written by M.S.W. and J.H., and L.C.L.H., H.J.v.d.W., S.W., and T.H. commented on the different manuscript versions.

## Competing interests

The authors declare no competing interests.
