## [Peer Review file · Nature Communications]

Loss of sea ice alters light spectra for aquatic photosynthesis

Corresponding Author: Professor Jef Huisman

Version 0:

Reviewer comments:

Reviewer #1

(Remarks to the Author)

This manuscript reports a detailed study of the differences in expected spectral light properties in the ocean beneath an ice cover and without an ice cover. Noteworthy results include the predicted spectral narrowing of the light field available for photosynthesis in open ocean with no ice. This narrowing includes a spectral signature with shorter wavelengths and more pronounced spectral niches. The results suggest that reduction of sea ice may likely lead to changes in pigment and taxonomic composition of primary producers in the polar ecosystem. The manuscript is clear, organized, very well written, and contains clear figures. The conclusions are well supported and the analysis and methodology is clear and straightforward. I recommend publication as is, after attention to the following very minor technical corrections.

replace "ice", with "sea ice" in many instances throughout the text

45: brine "inclusions" (more general than "channels"), several instances throughout manuscript. Also, not sure what is meant by "snow pockets"?

85: equation (singular), not equations

123: "(without snow cover and ice algae)", also without a surface scattering layer? (as defined in Reference 19)

165: "Superimposition" I think "superposition" instead

411: Add references to Fig 1c caption

Reviewer #2

(Remarks to the Author)

The manuscript titled, "Loss of sea ice alters light spectra for aquatic photosynthesis" by Soja-Woźniak et al., presents the idea that as climate change impacts sea ice cover in polar regions, there will be a loss of a light spectra-specific niches that could impact primary producer biodiversity. I note that my background in radiative transfer is not strong enough to comment on this aspect of the manuscript. Instead, I focus on the biological interpretation of the results. The concept is intriguing; however, the arguments and examples to explain how biodiversity could change are limited. I have provided some examples below relative to this statement.

(1) The comparison of change between under an ice cover to light available in the open ocean is really only comparable at the same intensity levels (i.e., all things being held equal, what is the change?). In the paragraph starting on line 265, Beeler Sooho (1987) is cited who compared P. antarctica between open water (ice edge) and under the ice. As noted in Beeler Soohoo et al. (1987), these sites were under completely different light intensities (open water vs ice covered) and spectra (due to absorption associated with congelation and platelet algal communities above the under-ice advected bloom) that led to their conclusions relative to changes in chl-specific absorption coefficients. In the submitted manuscript, the change in absorption near 500 nm from Beeler Soohoo et al. (1987) seems to be interpreted more relative to the physical radiative transfer differences that sea ice causes (i.e., smooths and shifts spectra relative to open water) rather than the impacts of overlying algal pigments and total irradiance on photoacclimation of an algal community between the open water and under-ice sites.

(2) In paragraph starting on Line 224, it is argued that the smoothed and shifted spectra in the ice and ice-covered

environment helps explain why diatoms and Phaeocystis dominate these environments. This statement was then refuted on line 258-259, "... diatoms and prymnesiophytes are not restricted to ice-covered waters...". In fact, they often dominate open water blooms when nutrients are replete. Similar to point (1) above, the interpretation seems to miss the impact of intra- (package effect) and inter- (influence of high algal concentrations in the ice environment) cellular self-shading that can cause photoacclimation and production of different pigments. Also, as rightfully pointed out in the discussion of the same paragraph, nutrients can greatly impact species composition, particularly in reference to diatom versus Phaeocystis dominance; however, the role of nutrients is largely glossed over in this paragraph.

As these arguments carry a large portion of the discussion, I believe this manuscript requires a bit of re-focus before being considered for publication.

Reviewer #3

(Remarks to the Author)

This study investigates the potential ramifications of the possible disappearance of summer sea ice in the Arctic and Antarctic on the light environment experienced by marine microalgae. The study places particular emphasis on the significance of "peaks" present in the absorption spectrum of pure seawater, which are notably absent in the absorption spectrum of pure sea ice. Through a series of radiative transfer simulations conducted in both sea ice and open ocean conditions, disparities are observed in the irradiance spectrum at the base of the euphotic zone.

While the abstract appeared promising, I found myself encountering several points of concern:

Firstly, sea ice in polar oceans typically exhibits low levels of CDOM. Simulations utilizing an absorption coefficient for CDOM reaching 20 m⁻¹ are notably unrealistic. Even in the Baltic Sea, which serves as an imperfect proxy for the Arctic or Antarctic Ocean, reported in-situ data suggests aCDOM barely exceeds 1 m⁻¹. Yet, a considerable portion of the conclusions drawn from the simulations hinge on substantial variations in aCDOM, surpassing the influence of differences in absorption between water and sea ice.

Secondly, the simulations depicted in figures 3 and 4 assume sea ice thicknesses of up to 30 m. However, the average thickness in the Arctic is approximately 1.5 m, and even less in the Antarctic. The rationale behind simulating such thick sea ice to compare equivalent optical depths appears questionable.

Thirdly, considering that nearly two-thirds of the Arctic Ocean is covered by first-year sea ice that melts annually, it prompts inquiry into whether significant distinctions exist between regions with perennial ice cover and those ice-free in summer. This warrants a thorough examination of existing data to discern any notable differences, rather than delving into discussions about cyanobacteria absent from polar oceans. Furthermore, it's worth noting that cryptophytes, which also possess phycobilins, are found at high latitudes.

Moreover, sea ice behaves somewhat akin to clouds, with its albedo primarily governing photon loss in a largely neutral manner within the visible spectrum. Consequently, the irradiance spectrum under sea ice broadly resembles the incident solar spectrum, save for the absorption of red light by the ice. Removing sea ice would elevate PAR at the surface without substantially altering the irradiance spectrum. This resemblance is likely more pronounced few meters below the surface. Hence, it appears that the primary change in surface irradiance resulting from sea ice removal is a general decrease, which warrants further consideration, at least for contextualizing the results.

Overall, I believe this study may overemphasize the impact of differences in the absorption spectrum of ice and seawater, exaggerate the influence of CDOM in ice through the use of excessively high values, simulate sea ice thicknesses that are unrealistic, and engage in speculative discourse regarding the potential impact of summer sea ice disappearance on spectral niches and microalgae selection in polar oceans, without sufficiently grounding its arguments in existing literature.

Therefore, I would cautiously advise against publication at this time.

Version 1:

Reviewer comments:

Reviewer #1

(Remarks to the Author)

The authors have adequately responded to my concerns and have made appropriate technical corrections. I believe this manuscript is now ready to be published as revised.

Reviewer #4

(Remarks to the Author)

This is the first time I am reviewing the paper that has already gone through a previous round of reviews. While I find the idea that there can be a shift in spectra between ice-covered and ice-free waters that would in turn impact primary producers, I do

not feel this manuscript is noteworthy enough for Nature Communications. My reasons are outlined below.

This is a modeling study that simulates light under sea ice and in the open ocean as a function of CDOM levels. I have several issues with the approach and its relevance:

1. Realistic sea ice is not used in this simulation. Sea ice is covered by snow until summer melt and plays the most important role in light entering the upper ocean under the sea ice. However, under low light levels, which can happen for thin snow, there is enough light already for ice algae to grow (see recent paper by Clara Hoppe in Nature using the MOSAiC data). How concentrations of ice algae influence then the light regime entering the in water column is important in the narrowing of the spectra yet is not discussed.
2. When looking at Figure 2 there is no discussion on what is a good fit of the model to observations. When there is a broader spectrum say for the Baltic Sea case, there is a pronounced shift between the modeled and observed spectra towards shorter wavelengths. How does this influence your results and interpretation? A shift is also seen in the Antarctic case.
3. Regarding Figure 3, while it is clear that at depth in the ocean, the model simulates a narrower distribution of light there does not appear to be a clear shift in the peak wavelength between the light under the ice and at depth in the water column. But besides this, I don't understand why this comparison is made. Instead the figure should show the same depths in the water column with and without sea ice, rather than showing the light under the ice and then state this doesn't compare to the light at depth in the water column. You are not comparing the same things. If you want to talk about sea ice vs. open water then the same comparison should be made, i.e. show the irradiance at depth in the water column with and without sea ice.
4. I do not understand the choice of ice albedos used, and why the Antarctic ice albedo is higher than that for the Arctic. It all depends on the thickness of the ice in the absence of snow but for 1.6 and 2m of ice, the bare ice albedo will not be different. 20% for 20cm ice may be realistic though it has to be backed by observations or a reference. The thickness of the SSL should also be specified.
5. I do not find that a sufficient sensitivity study has been performed as a function of ice characteristics to make this study significant. How sensitive are the results to the scattering coefficients used for example? There are many observations made by different studies that have different coefficients than those of Biegrib and Light. I realize extended Figure 3 tries to do a sensitivity study by assuming different thicknesses of the SSL, DL and ICE but that is not the same as considering the impact of the actual scattering coefficients used. These will be different for the different ice types.
6. It is difficult to make real implication statements as there are so many factors that influence primary productivity, including the initial blooms that happen under very low light levels, nutrient availability and timing of predation. Without real observations to support these statements it does feel a bit underwhelming.

Overall, I feel such a modeling study as this manuscript presents should be published somewhere, but to make it a Nature paper, I would have expected to see actual results on the taxonomy changes observed between a sea ice-covered regime and open water.

Version 2:

Reviewer comments:

Reviewer #5

(Remarks to the Author)

Review of manuscript 'Loss of sea ice alters light spectra for aquatic photosynthesis' by Woźniak et al. submitted to Nature Communications.

General comment

This is the first time that I review the paper Loss of sea ice alters light spectra for aquatic photosynthesis by Woźniak et al. The manuscript already went through two rounds of revisions and I was asked to evaluate the manuscript as well as the responses to Rev#4.

The paper is well written, the analysis properly conducted and the figures are of good quality. I find the results of the paper interesting and worth publishing, but I agree with Rev#4 that such study is not suited for Nature Communications. I will refer in the following to the general comments of Rev#4:

1. In response to Rev#4 the authors did include a sensitivity test accounting for the presence of snow, however there is no specification in the text on how the snow was included. As they showed, snow strongly attenuates light, however the effect of snow is wavelength-dependent (Mundy et al., 2007; Lange et al., 2016). I think this latter aspect is even more relevant for the present study rather than attenuation itself.
2. I find that the authors address this comment thoroughly
3. The authors compare spectra at the same optical depths in sea ice and open water, arguing on the comment of the reviewer to compare the spectra at the same depth in the water. The authors approach of comparing the spectra at optical depths is reasonable in order to assess the difference that the two environments (sea ice VS water) have on the light spectrum. However, for these findings to be relevant for phytoplankton communities in the water, I agree with Rev#4 that the authors should show the irradiance at the same depth in the water, with and without sea ice. I assume that the difference ice VS no-ice will be visible only at the surface where ice has the strongest effect. At depths larger than a couple of meters under the ice most likely the effect of the above water column will overcome any effect due to the absence of ice. This is an important aspect to then infer to which extent (to which depth) ecosystem may be affected by the disappearance of sea ice.
4. Addressed
5. Rebutted
6. The authors include in the discussion the point that light only, but also other factors (nutrients) do have effect on ocean

primary productivity. However, I agree that with lack of data this remains a purely modeling study, interesting and worth publication, but not in nature communications.

I also add that the authors should address comment 3 of Rev#4 to show the extent at which phytoplankton communities can really be affected by changes in the light spectrum due to sea-ice loss.

Detailed comments:

- L 61: it was discovered -> it has been shown

- L 115: remove 'sheet'

- L 126: Change 'Now that ...' in 'Once ...'

- L 202 euphotic depth (zone) is defined here but it is mentioned already above in the manuscript, so this definition should appear already before

- L 695: bice appears here for the first time, it needs to be defined in the text or at least here in the caption

References:

Mundy, C. J., J. K. Ehn, D. G. Barber, and C. Michel (2007), Influence of snow cover and algae on the spectral dependence of transmitted irradiance through Arctic landfast first-year sea ice, *J. Geophys. Res.*, 112, C03007, doi:10.1029/2006JC003683.

Lange, B. A., C. Katlein, M. Nicolaus, I. Peeken, and H. Flores (2016), Sea ice algae chlorophyll a concentrations derived from under-ice spectral radiation profiling platforms, *J. Geophys. Res. Oceans*, 121, 8511–8534, doi:10.1002/2016JC011991.

Response to the reviewer's comments

Response to Reviewer 1:

This manuscript reports a detailed study of the differences in expected spectral light properties in the ocean beneath an ice cover and without an ice cover. Noteworthy results include the predicted spectral narrowing of the light field available for photosynthesis in open ocean with no ice. This narrowing includes a spectral signature with shorter wavelengths and more pronounced spectral niches. The results suggest that reduction of sea ice may likely lead to changes in pigment and taxonomic composition of primary producers in the polar ecosystem. The manuscript is clear, organized, very well written, and contains clear figures. The conclusions are well supported and the analysis and methodology is clear and straightforward. I recommend publication as is, after attention to the following very minor technical corrections.

Response: We thank Reviewer #1 for the positive feedback and kind appreciation of our manuscript.

replace "ice", with "sea ice" in many instances throughout the text

Response: We have replaced "ice" by "sea ice" at more than 50 places in the manuscript.

45: brine "inclusions" (more general than "channels"), several instances throughout manuscript. Also, not sure what is meant by "snow pockets"?

Response: We changed "brine channels" to "brine inclusions" throughout the manuscript. Furthermore, we removed the "snow pockets" (line 47).

85: equation (singular), not equations

Response: We corrected "equations" to "equation" (singular), both on former line 85 (now line 88) and in the Methods section (line 334).

123: "(without snow cover and ice algae)", also without a surface scattering layer? (as defined in Reference 19)

Response: Good point; actually this was a very important comment. In our previous manuscript version, we had incorporated a depth-dependent scattering coefficient with a surface scattering layer in the model predictions for the central Arctic Ocean, Baltic Sea and McMurdo Sound Antarctica in Figure 2. However, we had ignored the surface scattering layer in our model calculations for Figures 3 and 4. Instead we had assumed a uniform scattering coefficient with a low value representative of the interior ice layer in Figures 3 and 4. (Our reasoning was that we wanted to keep the model as simple as possible). However, this assumption resulted in sea ice thicknesses of up to 30 m, which is unrealistic as Reviewer #3 correctly pointed out.

We have now included a surface scattering layer (SSL) in the model calculations for Figure 3 and 4, using the same approach as in Reference 19 (Smith et al. 2022; see also Briegleb & Light

2007). The values of the depth-dependent scattering coefficient in Figures 3 and 4 are based on Figure 8 of Light et al. (2015) and the values we applied in our model validation for the central Arctic Ocean, Baltic Sea and McMurdo Sound Antarctica (our Figure 2). The higher scattering coefficients in the top layers result in a much faster light attenuation. Hence, the ice thicknesses at the 5% and 1% light levels are now reached within a few meters, thus resolving the comment of Reviewer #3.

The addition of a surface scattering layer has led to the following changes in the manuscript: We have added a sentence in the Methods section (lines 337-340) and we added the new Extended Data Figure 4, to explain how depth-dependent scattering has been incorporated in our model. We provide the new parameter values for the depth-dependent scattering coefficient in Extended Data Table 2. All model calculations have been redone based on this new model structure, and all graphs in Figure 3, Figure 4 and Extended Data Figure 3 have been redrawn. We have updated the new values obtained from the model calculations in Table 1 and Extended Data Table 3. We have adjusted our descriptions of these figures and tables throughout the main text, in accordance with the new results. Finally, returning to the reviewer's comment, we now mention on former line 123 (now line 126) that the model includes a surface scattering layer.

165: "Superimposition" I think "superposition" instead

Response: We replaced "superimposition" by "overlay" (line 170 and line 175).

411: Add references to Fig 1c caption

Response: In the caption of Figure 1c we have now written "The scattering coefficient of sea ice varies depending on the amount of air bubbles and brine inclusions. Here we show a value of $b_{ice} = 20 \text{ m}^{-1}$, representative for the interior layer of Arctic sea ice", and added a reference to the paper of Bonnie Light et al. (2015; see their Figure 8).

Light, B., Perovich, D. K., Webster, M. A., Polashenski, C. & Dadic, R. Optical properties of melting first-year Arctic sea ice. Journal of Geophysical Research: Oceans 120, 7657–7675 (2015).

Response to Reviewer 2:

The manuscript titled, "Loss of sea ice alters light spectra for aquatic photosynthesis" by Soja-Woźniak et al., presents the idea that as climate change impacts sea ice cover in polar regions, there will be a loss of a light spectra-specific niches that could impact primary producer biodiversity. I note that my background in radiative transfer is not strong enough to comment on this aspect of the manuscript. Instead, I focus on the biological interpretation of the results. The concept is intriguing; however, the arguments and examples to explain how biodiversity could change are limited. I have provided some examples below relative to this statement.

Response: We thank Reviewer #2 for the kind and constructive comments on the biological interpretation of our results.

(1) The comparison of change between under an ice cover to light available in the open ocean is really only comparable at the same intensity levels (i.e., all things being held equal, what is the change?). In the paragraph starting on line 265, Beeler Sooho (1987) is cited who compared P.

antarctica between open water (ice edge) and under the ice. As noted in Beeler Soohoo et al. (1987), these sites were under completely different light intensities (open water vs ice covered) and spectra (due to absorption associated with congelation and platelet algal communities above the under-ice advected bloom) that led to their conclusions relative to changes in chl-specific absorption coefficients. In the submitted manuscript, the change in absorption near 500 nm from Beeler Soohoo et al. (1987) seems to be interpreted more relative to the physical radiative transfer differences that sea ice causes (i.e., smooths and shifts spectra relative to open water) rather than the impacts of overlying algal pigments and total irradiance on photoacclimation of an algal community between the open water and under-ice sites.

Response: This was an important comment for our revisions. Although it is always a struggle to keep Nature manuscripts short and to the point, we admit that our original description in this paragraph was too concise. We fully agree with the reviewer that the study of Beeler Soohoo et al. compared sites that differ both in light intensity and spectral composition. However, we could not find any field examples in the literature where absorption spectra of the same algal species were compared between open water and ice-covered water at the same light intensity (even though we fully agree with the reviewer that that would have been the ideal case to compare our model predictions with). Therefore, although their study does not provide the perfect comparison, we do think that the findings of Beeler Soohoo et al. (1987) are interesting, because of their detailed description of the irradiance spectra in and below sea ice and because of the observed difference in pigmentation between Phaeocystis below sea ice and in open water.

Therefore, we made the following changes in the manuscript:

- *In the first paragraph of the Introduction, we added a sentence to explain from the start that primary producers will be exposed to high light intensities when the ice melts, which can have major effects on photoacclimation and community composition (lines 34-37).*
- *In the Discussion, we now explain that - in the study of Beeler Soohoo et al. - the irradiance levels below sea ice were low and the spectrum was dominated by green wavelengths due to light attenuation by sea ice and self-shading by high densities of ice algae (lines 305-307).*
- *Furthermore, we now added in the Discussion that comparison of their findings with our model predictions is not straightforward. We have discussed the caveats as follows: "Yet, interpretation of these findings is not straightforward. Algae exposed to high irradiances near the water surface tend to have low contents of accessory photosynthetic pigments (including fucoxanthin) and high contents of photoprotective pigments to avoid photodamage (Robinson et al. 1997 ; Galindo et al. 2017; Lund-Hansen et al. 2020). Enhanced fucoxanthin contents are therefore often interpreted as a photoacclimative response to low irradiances, which may explain the high fucoxanthin contents under ice reported by Beeler Soohoo et al. (1987). At these low irradiances, however, acclimation to the prevailing light spectrum below sea ice likely also plays a role, as enhanced fucoxanthin contents clearly extend light harvesting for photosynthesis to longer wavelengths." (lines 315-323).*

Robinson, D. H., Kolber, Z., & Sullivan, C. W. (1997). Photophysiology and photoacclimation in surface sea ice algae from McMurdo Sound, Antarctica. Marine Ecology Progress Series 147: 243-256.

Galindo, V., Gosselin, M., Lavaud, J., Mundy, C. J., Else, B., Ehn, J., ... & Rysgaard, S. (2017). Pigment composition and photoprotection of Arctic sea ice algae during spring. Marine Ecology Progress Series 585: 49-69.

Lund-Hansen, L. C. et al. (2020). Effects of increased irradiance on biomass, photobiology, nutritional quality, and pigment composition of Arctic sea ice algae. Marine Ecology Progress Series 648:95–110.

(2) In paragraph starting on Line 224, it is argued that the smoothed and shifted spectra in the ice and ice-covered environment helps explain why diatoms and Phaeocystis dominate these environments. This statement was then refuted on line 258-259, "... diatoms and prymnesiophytes are not restricted to ice-covered waters...". In fact, they often dominate open water blooms when nutrients are replete.

Response: We fully agree with the reviewer that diatoms and prymnesiophytes dominate not only in ice-covered seas, but also in several other habitats (e.g., deeply mixed waters, turbid coastal waters). In our view, however, the statement that "diatoms dominate ice-covered environments" and the statement that "diatoms can also dominate other habitats" are not contradictory. Both statements are true. That is, diatoms are quite versatile and occupy several different ecological niches. However, triggered by a comment of Reviewer #3, that we should shift our focus to the phytoplankton of polar seas instead of cyanobacterial pigments, we have completely reorganized this part of the Discussion. Therefore, this paragraph has been largely rewritten (lines 285-301), and in doing so the statement that "... diatoms and prymnesiophytes are not restricted to ice-covered waters..." has been removed.

Similar to point (1) above, the interpretation seems to miss the impact of intra- (package effect) and inter- (influence of high algal concentrations in the ice environment) cellular self-shading that can cause photoacclimation and production of different pigments. Also, as rightfully pointed out in the discussion of the same paragraph, nutrients can greatly impact species composition, particularly in reference to diatom versus Phaeocystis dominance; however, the role of nutrients is largely glossed over in this paragraph.

Response: As outlined in our response to point (1) above, we now briefly discuss intercellular self-shading, photoacclimation and the production of different pigments in the context of the study of Beeler Soohoo et al. (1987) (lines 302-323). Furthermore, we are fully aware that nutrients can greatly impact species composition (see previous publications by several members of our team). Although nutrients are not the main topic of our current manuscript, we now discuss the role of nutrient limitation in the context of the observed shift from diatoms to Micromonas in the Canada Basin of the Arctic Ocean (e.g. Li et al. Science 2009; Tremblay et al. 2015) (see the new text on lines 286-291).

As these arguments carry a large portion of the discussion, I believe this manuscript requires a bit of re-focus before being considered for publication.

Response: We agree, and the above comments were very helpful to improve the ecological and physiological context of our manuscript.

Response to Reviewer 3:

This study investigates the potential ramifications of the possible disappearance of summer sea ice in the Arctic and Antarctic on the light environment experienced by marine microalgae. The study places particular emphasis on the significance of "peaks" present in the absorption spectrum of pure seawater, which are notably absent in the absorption spectrum of pure sea ice. Through a series of

radiative transfer simulations conducted in both sea ice and open ocean conditions, disparities are observed in the irradiance spectrum at the base of the euphotic zone.

While the abstract appeared promising, I found myself encountering several points of concern:

Firstly, sea ice in polar oceans typically exhibits low levels of CDOM. Simulations utilizing an absorption coefficient for CDOM reaching 20 m^{-1} are notably unrealistic. Even in the Baltic Sea, which serves as an imperfect proxy for the Arctic or Antarctic Ocean, reported in-situ data suggests aCDOM barely exceeds 1 m^{-1} . Yet, a considerable portion of the conclusions drawn from the simulations hinge on substantial variations in aCDOM, surpassing the influence of differences in absorption between water and sea ice.

Response: We agree with the reviewer that sea ice in polar oceans usually exhibits low levels of CDOM, and that $a\text{CDOM} = 20 \text{ m}^{-1}$ is indeed an extremely high value that is unrealistic for marine ecosystems. Nevertheless, the large Arctic rivers of Canada, Alaska and Russia flowing into the Arctic Ocean can easily reach aCDOM values of 1 to 5 m^{-1} , because of the enormous amounts of organic carbon stored in tundra peat layers. See for example the recent studies of Novak et al. (Frontiers in Marine Science 2022) and Juhls et al. (Remote Sensing of Environment 2022).

Our aim is to systematically investigate differences in irradiance spectra predicted for water versus sea ice, across a wide range of CDOM concentrations representative of different marine ecosystems. Therefore, we have now lowered the maximum CDOM value accordingly. In Figure 3 and Extended Data Figure 3, we have removed the panels with highest aCDOM of 10 m^{-1} , and now only show results for aCDOM = 0.01, 0.1 and 1 m^{-1} . Furthermore, we have adjusted the wide range of aCDOM values in Figure 4 by lowering the maximum value to 5 m^{-1} . We now mention in the legend of Figure 4, in the main text (lines 162-164) and in the Methods section (lines 404-405) that these high aCDOM values of up to 5 m^{-1} are representative for the river plumes of Arctic rivers, citing the studies of Novak et al. (2022) and Juhls et al. (2022). Although these changes affected our figures (Figure 3, Figure 4, and Extended Data Figure 3), they did not alter our main conclusions.

Novak MG, Mannino A, Clark JB, Hernes P, Tzortziou M, Spencer RGM, Kellerman AM and Grunert B (2022) Arctic biogeochemical and optical properties of dissolved organic matter across river to sea gradients. Front. Mar. Sci. 9: 949034. doi: 10.3389/fmars.2022.949034

Juhls B, Matsuoka A, Lizotte M, Bécu G, Overduin PP, El Kassir J, Devred E, Doxaran D, Ferland J, Forget MH, Hilborn A, Hieronymi M, Leymarie E, Maury J, Oziel L, Tisserand L, Anikina DOJ, Dillon M, Babin M (2022) Seasonal dynamics of dissolved organic matter in the Mackenzie Delta, Canadian Arctic waters: Implications for ocean colour remote sensing. Remote Sensing of Environment 283: 113327. doi: 10.1016/j.rse.2022.113327

Secondly, the simulations depicted in figures 3 and 4 assume sea ice thicknesses of up to 30 m. However, the average thickness in the Arctic is approximately 1.5 m, and even less in the Antarctic. The rationale behind simulating such thick sea ice to compare equivalent optical depths appears questionable.

Response: This is an important comment that has led to major changes throughout our manuscript. In Figures 3 and 4 (and also in Extended Data Figure 3), we had ignored the surface scattering layer. Instead we had assumed a uniform scattering coefficient with a low value representative of the interior ice layer. (Our reasoning was that we wanted to keep the model as simple as possible).

However, this assumption resulted in sea ice thicknesses of up to 30 m, which is unrealistic as Reviewer #3 correctly points out.

In line with a comment by Reviewer #1 (see above), we have now included a surface scattering layer (SSL) in the model calculations for Figure 3 and 4, using the same approach as in the models of e.g., Briegleb & Light (2007), Light et al. (2015) and Smith et al. (2022). The values of the depth-dependent scattering coefficient in Figures 3 and 4 are based on Figure 8 of Light et al. (2015) and the values we had applied in our model validation for the central Arctic Ocean, Baltic Sea and McMurdo Sound Antarctica (our Figure 2). The higher scattering coefficients in the top layers result in a much faster light attenuation. As a consequence, the ice thicknesses at the 5% and 1% light levels predicted by the model are now only a few meters or less, and hence within the realistic range (see the new versions of Figures 3 and 4).

The addition of a surface scattering layer has led to the following changes in the manuscript: We have added a sentence in the Methods section (lines 337-340) and we added the new Extended Data Figure 4, to explain how depth-dependent scattering has been incorporated in our model. We provide the new parameter values for the depth-dependent scattering coefficient in Extended Data Table 2. All model calculations have been redone based on this new model structure, and all graphs in Figure 3, Figure 4 and Extended Data Figure 3 have been redrawn. We have updated the new values obtained from the model calculations in Table 1 and Extended Data Table 3. Finally, we have adjusted our descriptions of these figures and tables throughout the main text, in accordance with the new results.

Briegleb BP and Light B (2007) A delta-Eddington multiple scattering parameterization for solar radiation in the sea ice component of the Community Climate System Model. NCAR Tech. Note TN-4721STR, 100 pp, National Center for Atmospheric Research, Boulder, Colorado.

Light B, Perovich DK, Webster MA, Polashenski C, Dadic R (2015). Optical properties of melting first-year Arctic sea ice. *Journal of Geophysical Research: Oceans* 120: 7657-7675. doi:10.1002/2015JC011163

Smith MM, Light B, Macfarlane AR, Perovich DK, Holland MM, Shupe MD (2022) Sensitivity of the Arctic sea ice cover to the summer surface scattering layer. *Geophysical Research Letters* 49: e2022GL098349. doi:10.1029/2022GL098349

Thirdly, considering that nearly two-thirds of the Arctic Ocean is covered by first-year sea ice that melts annually, it prompts inquiry into whether significant distinctions exist between regions with perennial ice cover and those ice-free in summer.

Response: We briefly touch upon optical properties of different types of sea ice in the Discussion (lines 216-226). In particular, we now compare sea ice with a weak surface scattering layer (SSL) and low scattering coefficients against sea ice with a strong SSL and high scattering coefficients. This comparison is somewhat akin to a comparison between first-year ice and multi-year ice, although we want to be cautious in our wording here as other differences between ice types (e.g., ponded ice versus snow-covered ice) have a stronger effect on the optical properties of ice than age per se. The results are shown in the new version of Extended Data Figure 3. In the Discussion, we have now added: “The results show that sea ice with a weak surface scattering layer and low scattering coefficients will have higher light transmission but slightly narrower irradiance spectra than sea ice with a strong surface scattering layer” (lines 221-224). We agree that a more extensive comparison

of different ice types would also be interesting, but that would require a novel study with new data and new model analyses, which is beyond the scope of our current manuscript.

This warrants a thorough examination of existing data to discern any notable differences, rather than delving into discussions about cyanobacteria absent from polar oceans. Furthermore, it's worth noting that cryptophytes, which also possess phycobilins, are found at high latitudes.

Response: This was a key comment. We agree that cyanobacteria are essentially absent from the polar oceans. Hence, we have removed the absorption spectra of cyanobacteria from Figure 5b, and replaced them by absorption spectra of phytoplankton taxa common in polar seas (prasinophytes and cryptophytes). See the new version of Figure 5b. We have completely rewritten the accompanying texts in the Discussion, which span four paragraphs (lines 239-301).

*Specifically, we now explain that the small prasinophyte *Micromonas polaris* is one of the most abundant planktonic algae of the oligotrophic Arctic Ocean, that it has a narrower absorption spectrum than diatoms, that its main photosynthetic pigments chlorophyll a and chlorophyll b capture the violet and blue niche, and that it has strongly increased in abundance at the cost of diatoms in the Canada Basin (e.g., Li et al. Science 2009) where ice cover has diminished considerably in recent decades (e.g., Babb et al. 2022). We also briefly describe the diversity of phycobilins in cryptophytes and the wavelengths they absorb (Greenwold et al. 2019), and the increasing abundance of cryptophytes in coastal waters of the West Antarctic Peninsula (e.g., Mendes et al. 2023). Furthermore, we explain that our model suggests that the loss of ice cover will cause a decrease in the diversity of cryptophyte pigment types, which is in line with the observation that the diversity of cryptophytes in these coastal waters is surprisingly low (Brown et al. 2021).*

Of course, we are fully aware that many other factors affect the species composition of phytoplankton communities as well. Phytoplankton in the polar seas have to adjust to many changes occurring simultaneously, including changes in light quality and quantity, nutrient conditions, stratification, and many other biotic and abiotic variables. Nevertheless, the overall correspondence between changes in irradiance spectra due to the loss of sea ice and changes in pigmentation of ice algae versus open ocean phytoplankton is remarkable, indicating that light quality is an important selective factor in communities of photosynthetic organisms.

We thank the reviewer for raising this comment, and we believe that these major changes in the Discussion have strongly increased the ecological relevance of our study.

Li, W. K. W., McLaughlin, F. A., Lovejoy, C. & Carmack, E. C. (2009). Smallest algae thrive as the Arctic Ocean freshens. Science 326: 539.

Babb, D. G. et al. (2022). Increasing multiyear sea ice loss in the Beaufort Sea: a new export pathway for the diminishing multiyear ice cover of the Arctic Ocean. Geophysical Research Letters 49, e2021GL097595.

Greenwold, M. J., Cunningham, B. R., Lachenmyer, E. M., Pullman, J. M., Richardson, T. L. & Dudycha, J. L. (2019). Diversification of light capture ability was accompanied by the evolution of phycobiliproteins in cryptophyte algae. Proc. R. Soc. London B 286: 20190655.

Mendes, C. R. B., Costa, R. R., Ferreira, A., Jesus, B., Tavano, V. M., Dotto, T. S., Leal, M. C., Kerr, R., Islabão, C. A., Franco, A. O. R., Mata, M. M., Garcia, C. A. E. & Secchi, E. R. (2023). Cryptophytes: an emerging algal group in the rapidly changing Antarctic Peninsula marine environments. Global Change Biology 29: 1791-1808.

Brown, M. S. et al. (2021). Low diversity of a key phytoplankton group along the West Antarctic Peninsula. Limnology and Oceanography 66, 2470–2480.

Moreover, sea ice behaves somewhat akin to clouds, with its albedo primarily governing photon loss in a largely neutral manner within the visible spectrum. Consequently, the irradiance spectrum under sea ice broadly resembles the incident solar spectrum, save for the absorption of red light by the ice. Removing sea ice would elevate PAR at the surface without substantially altering the irradiance spectrum. This resemblance is likely more pronounced few meters below the surface. Hence, it appears that the primary change in surface irradiance resulting from sea ice removal is a general decrease, which warrants further consideration, at least for contextualizing the results.

Response: We agree that the high albedo of sea ice removes a large fraction of the incident solar radiation in a wavelength-independent manner. Furthermore, light attenuation in sea ice is dominated by scattering due to air bubbles and brine inclusions, which is also essentially wavelength independent, while absorption by ice makes a much smaller contribution to light attenuation and is highest in the red part of the spectrum. These aspects are all included in our model, illustrated in Figure 1, and also highlighted in the first paragraph of the Discussion to explain why irradiance spectra below ice span a wide range of wavelengths. We have now added to this paragraph that irradiance spectra below sea ice “broadly resemble the incident solar spectrum” (lines 196-197).

We also agree that ice algae and phytoplankton living in or below the ice will be exposed to high surface irradiances when the ice cover disappears. Therefore, in the first paragraph of the Introduction, we have now added a sentence to explain from the start that primary producers will be exposed to high surface irradiance when the ice melts. Specifically, we have now written: “The high light intensities in the surface layer to which these primary producers will be exposed when the ice melts can have major effects on photoacclimation and community composition, and the implications of increased irradiance have therefore been extensively investigated (Arrigo et al. 2008; Galindo et al. 2017; Lund-Hansen et al. 2020) (lines 34-37).

Furthermore, we come back to this point in the Discussion, where we explain that algae exposed to high irradiances near the water surface tend to have low contents of accessory photosynthetic pigments (including fucoxanthin) and high contents of photoprotective pigments to avoid photodamage (Robinson et al. 1997; Galindo et al. 2017; Lund-Hansen et al. 2020) (lines 316-318).

Robinson, D. H., Kolber, Z., & Sullivan, C. W. (1997). Photophysiology and photoacclimation in surface sea ice algae from McMurdo Sound, Antarctica. Marine Ecology Progress Series 147: 243-256.

Arrigo, K. R., van Dijken, G. & Pabi, S. Impact of a shrinking Arctic ice cover on marine primary production. Geophysical Research Letters 35, 2008GL035028 (2008).

Galindo, V., Gosselin, M., Lavaud, J., Mundy, C. J., Else, B., Ehn, J., ... & Rysgaard, S. (2017). Pigment composition and photoprotection of Arctic sea ice algae during spring. Marine Ecology Progress Series 585: 49-69.

Lund-Hansen, L. C. et al. (2020). Effects of increased irradiance on biomass, photobiology, nutritional quality, and pigment composition of Arctic sea ice algae. Marine Ecology Progress Series 648:95–110.

Overall, I believe this study may overemphasize the impact of differences in the absorption spectrum of ice and seawater, exaggerate the influence of CDOM in ice through the use of excessively high values, simulate sea ice thicknesses that are unrealistic, and engage in speculative discourse regarding the potential impact of summer sea ice disappearance on spectral niches and microalgae selection in polar oceans, without sufficiently grounding its arguments in existing literature.

Response: We thank the reviewer for the thoughtful comments. We have now written more explicitly in the Abstract that light attenuation in sea ice is dominated by scattering (line 19) and we have increased the contribution of scattering in the model by adding a surface scattering layer. Furthermore, we have lowered the maximum CDOM concentration by a factor four to values representative of the river plumes of Arctic rivers, the sea ice thicknesses simulated by the model are now in the realistic range (i.e., a few decimeters to a few meters), and we have rewritten large parts of the Discussion to better connect our results to changes in microalgal composition that are currently being observed in Arctic and Antarctic waters. All these changes are firmly based upon the existing literature, which has increased the number of references (from 55 references in our original submission to 73 references in the revised manuscript).

Our own changes:

1. We shortened the Abstract to 150 words, in line with the formatting instructions of Nature Communications.
2. In our original submission, we argued in the Discussion that the results did not show a spectral niche in the violet region (< 450 nm) in Figure 4d, because the low solar elevation in polar regions largely removed the shortest wavelengths by Rayleigh scattering in the atmosphere. However, this explanation was not correct. Of course, the sun does have a low elevation in polar regions, but this was not the explanation for the lack of the violet wavelengths. Instead, we had inadvertently applied a relatively high absorption coefficient for non-algal particles ($a_{\text{NAP}(440)} = 0.01 \text{ m}^{-1}$), which effectively removed the violet wavelengths even at low CDOM concentrations. We have now used a lower absorption coefficient for non-algal particles of $a_{\text{NAP}(440)} = 0.001 \text{ m}^{-1}$, which is representative for clear ocean waters in the Arctic (Matsuoka et al. 2011).

As a consequence, the results now also show a spectral niche in the violet region of the spectrum (see the new Figure 4d). The reference to Matsuoka et al. (2011) has been added in the Methods section (Ref 66 on line 413) and the new value of $a_{\text{NAP}(440)} = 0.001 \text{ m}^{-1}$ is implemented in Extended Data Table 2.

Matsuoka, A., Hill, V., Huot, Y., Babin, M. & Bricaud, A. (2011). Seasonal variability in the light absorption properties of western Arctic waters: parameterization of the individual components of absorption for ocean color applications. *Journal of Geophysical Research: Oceans* 116: C02007.

3. We stored all data of our figures in figshare and added a Data Availability statement.
4. As explained in the Methods, all our model calculations were done with the commercial software package Hydrolight-Ecolight version HE6.0 (lines 332-334), which is a state-of-the-art radiative transfer model that solves the radiative transfer equation numerically. The parameter settings for these calculations are summarized in Extended Data Tables 1 and 2. Hence, we did not write our own code, and therefore did not add a Code Availability statement.

Response to the reviewer's comments

Response to Reviewer 1:

The authors have adequately responded to my concerns and have made appropriate technical corrections. I believe this manuscript is now ready to be published as revised.

Response: We thank Reviewer #1 for the positive feedback and kind appreciation of our manuscript.

Response to Reviewer 4:

This is the first time I am reviewing the paper that has already gone through a previous round of reviews. While I find the idea that there can be a shift in spectra between ice-covered and ice-free waters that would in turn impact primary producers, I do not feel this manuscript is noteworthy enough for Nature Communications. My reasons are outlined below.

Response: We thank the reviewer for the time invested in our manuscript. We are pleased to read that the reviewer appreciates the idea that there can be a shift in spectra between ice-covered and ice-free waters that would in turn impact primary producers. Obviously, we do feel that our novel findings are noteworthy for Nature Communications and we also think the reviewer misinterpreted some of our results, as we will outline in our responses below. We hope our explanations below as well as the changes and new analyses in the manuscript will clarify most of the issues.

This is a modeling study that simulates light under sea ice and in the open ocean as a function of CDOM levels. I have several issues with the approach and its relevance:

1. Realistic sea ice is not used in this simulation. Sea ice is covered by snow until summer melt and plays the most important role in light entering the upper ocean under the sea ice. However, under low light levels, which can happen for thin snow, there is enough light already for ice algae to grow (see recent paper by Clara Hoppe in Nature using the MOSAiC data). How concentrations of ice algae influence then the light regime entering the in water column is important in the narrowing of the spectra yet is not discussed.

Response: Indeed, our model considers sea ice in late spring and summer, when the snow cover has melted and a surface scattering layer develops due to ablation of the top layer of sea ice. The seasonal decrease in snow cover and subsequent development of the surface scattering layer are nicely illustrated in Figure 1 of Macfarlane et al. (Elementa Science of the Anthropocene 2023), which is also based on the MOSAiC expedition. In our view, investigating how the spectra are affected by the loss of sea ice in the summer season is a perfectly valid scenario, especially because the loss of sea ice is most dramatic towards the end of the summer period. We now explain the seasonal setting of our model at the start of the Results section (lines 87-89): "We consider bare sea ice, representative of late spring and summer when the snow has melted. After snow melt, often a porous granular structure develops in the top layer of melting sea ice known as the surface scattering layer (Macfarlane et al. 2023)."

We fully agree that in spring, the ice is often covered by a layer of snow, which can strongly diminish light levels below sea ice. It was interesting to read the recent findings of Hoppe et al. (2024) that the very low light levels below snow-covered ice can already provide sufficient irradiance for algae to grow. Incidentally, we note that the paper of Hoppe et al. (2024) was published on 4 September 2024, i.e., after we had submitted our revised manuscript to the journal. Hence, their findings were not available to us earlier.

In response to these comments, we added a new sensitivity analysis in which we compared irradiance spectra for different degrees of snow cover and extended our analysis to lower light levels (see the new Extended Data Figure 5). The results are described on lines 180-187:

“Furthermore, in early spring, sea ice is often covered by a layer of snow (Ardyna & Arrigo 2020, Macfarlane et al. 2023), which has a very high albedo and can strongly diminish light levels (Perovich 2007, Lund-Hansen et al. 2014). Our calculations revealed that even a thin layer of only 3 cm of snow can remove almost 90% of the incident irradiance, in line with field observations (Hamre 2004; Lund-Hansen et al. 2014). Yet, the low light levels below snow-covered ice can already provide sufficient irradiance for ice algae to grow (Hancke et al. 2018; Hoppe et al. 2024). We therefore also compared irradiance spectra below ice for different degrees of snow cover, and extended the comparison to lower light levels. The results show that sea ice covered by snow will generate even broader irradiance spectra than bare sea ice without snow cover (Extended Data Figure 5).”

Our model does include light absorption and scattering by ice algae and phytoplankton. See equations (3) and (4) in the Methods section. Their effect on the light regime depends on their concentration. For example, in our case study of the Central Arctic Ocean, the chlorophyll a concentration in sea ice is very low and has hardly any effect on the calculated spectrum (Figure 2a). However, the chlorophyll a concentrations of ice algae were higher at the Baltic Sea (Figure 2b) and very high at McMurdo Sound in Antarctica (Figure 2c), where it had an effect on the calculated spectra. How these ice algae are important in narrowing the spectra entering the water column is explicitly discussed in the Results section. On lines 115-119, we wrote:

“At McMurdo Sound, in Antarctica the chlorophyll a concentration of ice algae was very high (15.2 mg m⁻²). Hence, both violet-blue and red wavelengths were largely absorbed by chlorophyll, and the light spectrum underneath the ice cover was dominated by green and yellow wavelengths.”

In the next section of the Results, entitled ‘Comparison of light spectra below sea ice and in open water’, we model light regimes without ice algae or phytoplankton. We do this on purpose, to predict irradiance spectra generated by the abiotic environmental conditions, because these spectra provide the template for new algal photosynthesis and growth. We now explain this at the beginning of this section: “The concentration of ice algae and phytoplankton is set to zero to predict the light spectra generated by the abiotic environment (i.e., by sea ice, seawater, CDOM and NAP). In other words, we calculate the spectra that will be available for aquatic photosynthesis when new populations of ice algae or phytoplankton start to grow.” (lines 130-134).

2. When looking at Figure 2 there is no discussion on what is a good fit of the model to observations. When there is a broader spectrum say for the Baltic Sea case, there is a pronounced shift between the modeled and observed spectra towards shorter wavelengths. How does this influence your results and interpretation? A shift is also seen in the Antarctic case.

In response to this comment, we made several changes in our model simulations for Figure 2:

- In our original submission we used a 4-layered system of ice to model the irradiance spectra in Figure 2. We now changed it to a 3-layered system (SSL, DL, INT) in order to make the

approach fully consistent throughout the manuscript, since we used a 3-layered system for the model calculations in all other figures. The thicknesses of the different layers (SSL, DL, INT) are specified in Extended Data Table 1.

- *We now adopted the empirical relation of Babin et al. (2003) to convert the absorption coefficient of NAP ($a_{\text{NAP}}(440)$) to the NAP concentration (see the new sentence added on line 446-447); this affected the NAP concentration used to calculate the spectral scattering by NAP (see Extended Data Table 1).*
- *The Baltic Sea study of Kari et al. (2020) measured SPM concentrations for both sea ice and seawater, and we now use these data to distinguish between $a_{\text{NAP}}(440)$ for sea ice versus seawater in the Baltic Sea (see Extended Data Table 1).*
- *Our co-author Dr. Lars C. Lund-Hansen provided unpublished data of the albedo (65%), ice thickness (198 cm) and chl-a concentration in water (1.1 mg/m³) at McMurdo Sound, Antarctica, which he had measured on location simultaneously with the irradiance spectrum in Figure 2c. We now incorporated these data in the model (see Extended Data Table 1).*
- *After this new parameterization, we re-estimated the value of $a_{\text{CDOM}}(440)$ at McMurdo Sound and the values of the scattering coefficients at all three locations by a least-squares fit of the model to the measured spectra (see Extended Data Table 1).*

Individually, the effects of each of these changes were small, but together they helped to further improve the model fits (see new Figure 2). There are still slight differences between the modeled and observed spectra, but they are not systematic and they are less pronounced.

We now added the coefficient of determination (R^2 value) in the new Figure 2 to quantify the fit of the model to the observed spectrum. The R^2 values of the Arctic, Baltic and Antarctic ranged from 0.93 to 0.99 (which is pretty impressive in our view!). Furthermore, we now added in the main text: “Probably, the model predictions could be even further improved by fine-tuning the parameter values (e.g., by small changes in CDOM or chlorophyll concentration), but we preferred to base our model predictions as much as possible on measured parameter values (Extended Data Table 1).” (lines 121-124)

Hence, the relatively small differences between modeled and observed spectra in Figure 2 are essentially a parameterization issue; they are very different from the systematic differences between the spectra generated by sea ice versus open ocean water reported in Figures 3 and 4.

3. Regarding Figure 3, while it is clear that at depth in the ocean, the model simulates a narrower distribution of light there does not appear to be a clear shift in the peak wavelength between the light under the ice and at depth in the water column. But besides this, I don't understand why this comparison is made. Instead the figure should show the same depths in the water column with and without sea ice, rather than showing the light under the ice and then state this doesn't compare to the light at depth in the water column. You are not comparing the same things. If you want to talk about sea ice vs. open water then the same comparison should be made, i.e. show the irradiance at depth in the water column with and without sea ice.

Response: The full width at half maximum (FWHM), mean wavelength and peak wavelength of the irradiance spectra in Figure 3 are reported in Table 1. We agree that the shifts in mean wavelength of the spectra below sea ice versus open water are more pronounced than the shifts in peak wavelength. Still, there are clear shifts in peak wavelength, although they are better visible at the 25% optical depth than in the tiny spectra at the 1% optical depth. For instance, at the 25% optical depth, the peak wavelength of the oligotrophic ocean shifts from 533 nm below ice to 478 nm in open water. Likewise, the peak wavelength shifts from 548 nm to 498 nm for the mesotrophic ocean, and from 608 nm to 583 nm for coastal waters (see Table 1). These are substantial changes!

We now inverted Table 1, by showing the properties of the spectra below sea ice and in open water in the columns of the table and the different ocean ecosystems in the rows of the table. This makes it much easier to read the table. Furthermore, in the text, we now describe the shift in the mean wavelength of the spectra instead of the peak wavelength, because the spectral shifts are indeed more pronounced for the mean wavelength. That is, on lines 157-159, we now write: “For example, at an optical depth of 25%, the mean wavelength of the irradiance spectrum in the oligotrophic ocean is shifted from the green part of the spectrum (586 nm) below ice to the blue part of the spectrum (486 nm) in open ocean water (Figure 3a,d; Table 1).” This is a shift of 100 nm!

We compare spectra at the same optical depths, not at the same physical depths. This is a standard approach commonly used for the comparison of different spectra. See e.g., the Wikipedia pages on optical depth or the textbooks on marine and atmospheric optics. The advantage of comparing spectra at the same optical depth is that the shape of the spectra can be compared while the total irradiance integrated over the full spectra (their ‘total light intensity’) is the same. This was also emphasized by Reviewer #2 in the previous round of review, who wrote: “The comparison of change between under an ice cover to light available in the open ocean is really only comparable at the same intensity levels”. Hence, we are “comparing the same things”.

We now added a sentence in the Results section to explain this more clearly (lines 127-130): “The comparisons are made at the same optical depth, i.e., at the same total irradiance integrated over the spectrum. In this way, the shape of the spectra can be compared without potentially confounding effects of differences in total irradiance.” Furthermore, we changed the term “photic depth” to “optical depth” throughout the manuscript, because the latter term is more widely used.

4. I do not understand the choice of ice albedos used, and why the Antarctic ice albedo is higher than that for the Arctic. It all depends on the thickness of the ice in the absence of snow but for 1.6 and 2m of ice, the bare ice albedo will not be different. 20% for 20cm ice may be realistic though it has to be backed by observations or a reference. The thickness of the SSL should also be specified.

Response: Good point. The value that we used for the albedo of the Central Arctic Ocean is based on measurements of Lund-Hansen et al. (2015). Unfortunately, we made an error in Extended Data Table 1, for which we apologize. The albedo for the Central Arctic Ocean should indeed not be 40%. Lund-Hansen et al. (2015) measured an albedo value of 55% and that is the value that we have used in our model simulations. Hence, we have now corrected the albedo for the Central Arctic Ocean to 55% in Extended Data Table 1.

Furthermore, the albedo of 60% for McMurdo Sound, Antarctica, indicated in our previous manuscript version was merely an estimate. We have now updated this with actual measurements of the albedo at McMurdo Sound, Antarctica by our co-author Lars Lund-Hansen (unpublished data), measured on location simultaneously with the irradiance spectrum of Figure 2c. These measurements gave an Antarctic ice albedo of 65%, which we have now incorporated in our model simulations and Extended Data Table 1.

The albedo of 21% for the Baltic Sea was based on reflectance spectra of bare ice in Figure 6b of Kari et al. (2020), where we used the lowest value of the range. We now added a reference to Kari et al. (2020) for the albedo of the Baltic Sea in Extended Data Table 1.

The thicknesses of the different layers (SSL, DL, INT) are specified in Extended Data Table 1.

5. I do not find that a sufficient sensitivity study has been performed as a function of ice characteristics to make this study significant. How sensitive are the results to the scattering coefficients used for example? There are many observations made by different studies that have different coefficients than those of Biegrib and Light. I realize extended Figure 3 tries to do a

sensitivity study by assuming different thicknesses of the SSL, DL and ICE but that is not the same as considering the impact of the actual scattering coefficients used. These will be different for the different ice types.

Response: This comment is not correct. For some reason, the reviewer believes that the sensitivity analysis in our Extended Data Figure 4 (which was Extended Data Figure 3 in the previous manuscript version) assumed different thicknesses of the SSL, DL and ICE. However, this was not the case. Our Extended Data Figure 4 investigated the sensitivity of the model predictions to different values of the scattering coefficients of ice (b_{ice}). Therefore, we already performed the sensitivity analysis requested by the reviewer!

We have ensured that Extended Data Figure 4 clearly states that it investigates the scattering coefficients. In the panels of Extended Data Figure 4 (our former Ext Data Figure 3), we specify the different values of b_{ice} (which has units of m^{-1}). Furthermore, the caption of Extended Data Figure 4 explicitly states “Sensitivity of the model predictions to the scattering coefficient of ice (b_{ice}).”

Furthermore, we have now conducted additional sensitivity analysis. We added a sensitivity analysis of the albedo of ice in the new Extended Data Figure 3. And in response to comment #1 above, we added a sensitivity analysis of the effect of snow cover in the new Extended Data Figure 5.

We moved the sensitivity analysis from the Discussion to a new section in the Results called ‘Variation in optical properties of sea ice’, where we present and discuss the results of these new sensitivity analyses (lines 171-190).

6. It is difficult to make real implication statements as there are so many factors that influence primary productivity, including the initial blooms that happen under very low light levels, nutrient availability and timing of predation. Without real observations to support these statements it does feel a bit underwhelming.

Overall, I feel such a modeling study as this manuscript presents should be published somewhere, but to make it a Nature paper, I would have expected to see actual results on the taxonomy changes observed between a sea ice-covered regime and open water.

Response: We fully agree that primary production is affected by many factors, including light levels, nutrient availability, predation, and so on. Hence, the reviewer is preaching to the choir 😊. We have also extensively published about effects of light, nutrients, grazing, viruses, etc, on primary production ourselves (see previous publications by several of our team members).

*Our present manuscript is indeed a model study. It presents new insights on how the loss of ice cover will affect irradiance spectra for aquatic photosynthesis according to current state-of-the-art models in marine optics. However, we do compare our model predictions with real observations. Specifically, the light spectra predicted by our model are compared with light spectra measured in the Arctic Ocean, Baltic Sea, and at Antarctica (Figure 2). Furthermore, we extensively compare our model findings with major taxonomic changes observed in polar waters by other studies, such as the shift from diatoms to *Micromonas* spp in the Canada Basin of the Arctic Ocean (lines 283-293) and the increase of cryptophytes in coastal waters of the West Antarctic Peninsula (lines 294-307).*

It is clearly beyond the scope of our current manuscript to organize a series of polar expeditions to collect new data on taxonomic changes of ice algae and phytoplankton. However, we do hope our novel findings will inspire new polar expeditions, to investigate how the loss of sea ice will affect marine primary producers and to collect new data linking irradiance spectra with changes in algal pigmentation and taxonomy in Arctic and Antarctic waters.

Response to the reviewer's comments

Response to Reviewer 5:

General comment

This is the first time that I review the paper Loss of sea ice alters light spectra for aquatic photosynthesis by Woźniak et al. The manuscript already went through two rounds of revisions and I was asked to evaluate the manuscript as well as the responses to Rev#4. The paper is well written, the analysis properly conducted and the figures are of good quality. I find the results of the paper interesting and worth publishing, but I agree with Rev#4 that such study is not suited for Nature Communications.

Response: We thank Reviewer #5 for the kind words of appreciation for our manuscript, pointing out that the paper is well written, the analysis properly conducted, and the figures are of good quality. We are also most grateful for the reviewer's comments. Obviously, we disagree on one point, in the sense that we do feel that our novel findings are highly suited for Nature Communications (see our reply to point 6 below). However, we appreciate the reviewer's constructive approach that has helped us to further improve (and hopefully finalize) the manuscript.

I will refer in the following to the general comments of Rev#4:

1. In response to Rev#4 the authors did include a sensitivity test accounting for the presence of snow, however there is no specification in the text on how the snow was included. As they showed, snow strongly attenuates light, however the effect of snow is wavelength-dependent (Mundy et al., 2007; Lange et al., 2016). I think this latter aspect is even more relevant for the present study rather than attenuation itself.

Response: We are well aware that light attenuation by snow is wavelength-dependent, and this was incorporated in our model. In the caption of Extended Data Figure 5, in the previous version of our manuscript, we wrote that "The albedo and spectral extinction coefficient of snow are based on Figures 3 and 5 of Perovich (2007)". Figure 5 of Perovich (2007) shows the wavelength-dependence of the extinction coefficient of snow; it is very similar to Figure 4 of Mundy et al. (2007). However, the data of Perovich (2007) spanned the full spectral range from 400 - 850 nm, while Mundy et al. (2007) covered only the range from 400 - 700 nm. Hence, we used the data of Perovich (2007) in our model.

We have now added a short paragraph in the Methods section to explain how snow was included in the model (lines 492-501). Furthermore, in the main text, we now state explicitly that light attenuation by snow is wavelength-dependent, with references to both Perovich et al. (2007) and Mundy et al. (2007) (ref 37 and 38) (lines 204-205).

Perovich, D. K. Light reflection and transmission by a temperate snow cover. Journal of Glaciology 53, 201–210 (2007).

Mundy, C. J., Ehn, J. K., Barber, D. G. & Michel, C. Influence of snow cover and algae on the spectral dependence of transmitted irradiance through Arctic landfast first-year sea ice. Journal of Geophysical Research 112, C03007 (2007).

2. I find that the authors address this comment thoroughly

Response: Thank you!

3. The authors compare spectra at the same optical depths in sea ice and open water, arguing on the comment of the reviewer to compare the spectra at the same depth in the water. The authors approach of comparing the spectra at optical depths is reasonable in order to assess the difference that the two environments (sea ice VS water) have on the light spectrum. However, for these findings to be relevant for phytoplankton communities in the water, I agree with Rev#4 that the authors should show the irradiance at the same depth in the water, with and without sea ice. I assume that the difference ice VS no-ice will be visible only at the surface where ice has the strongest effect. At depths larger than a couple of meters under the ice most likely the effect of the above water column will overcome any effect due to the absence of ice. This is an important aspect to then infer to which extent (to which depth) ecosystem may be affected by the disappearance of sea ice.

Response: We have added a new Figure to address this comment (Figure 3). The first two panels of this figure show scalar irradiance spectra at the same depths in the water column, with and without sea ice (Figure 3a,b). In line with expectation, the results show that irradiance in water below sea ice is much lower than irradiance at the same depths in open ocean water. The model predictions also suggest differences in the spectral composition of ice-covered versus open ocean water, although this is difficult to assess in view of the concomitant differences in total irradiance. The new Figure 3a,b is described on lines 137-146.

For comparison, we have also added two other panels in the new Figure 3, which show scalar irradiance spectra at the same optical depths (Figure 3c,d). In this way, the shape of the spectra in the euphotic zone can be compared without potentially confounding effects of differences in total irradiance. Moreover, this visualization also helps to address the second part of the reviewer's comment. At the 10% and 5% optical depth (i.e., at 4.6 m and 16.5 m depth), water below sea ice still has a relatively broad irradiance spectrum that covers violet, blue and green wavelengths (400-600 nm; Figure 3c), whereas open ocean water at the same optical depths is characterized by a more pronounced peak largely confined to the violet and blue part of the spectrum (400-500 nm; Figure 3d). However, deeper down in the euphotic zone, at 2% and 1% optical depth, the water column has filtered out the longer wavelengths and the spectra below sea ice and in open ocean water become more similar. Hence, for the clear ocean water used in this example, the effect of sea ice on the underwater spectrum extends to at least 16.5 m depth but gradually vanishes at greater depths. The new Figure 3c,d is described on lines 147-166.

4. Addressed

Response: Thank you!

5. Rebutted

Response: Thank you!

6. The authors include in the discussion the point that light only, but also other factors (nutrients) do have effect on ocean primary productivity. However, I agree that with lack of data this remains a purely modeling study, interesting and worth publication, but not in nature communications.

Response: We agree that our manuscript is largely a model study. To our knowledge, Nature Communications and many other journals in the Nature family regularly publish model studies, and

the development of new models has contributed to major new scientific insights and discoveries in, e.g., physics, chemistry, astronomy, earth sciences, biology, and many other fields of science.

*We emphasize that our model study is firmly based on data. The model assumptions and parameter estimates are based on real-world observations (see Extended Data Tables 1 and 2). The predicted spectra are compared with light spectra measured in the Arctic Ocean, Baltic Sea, and at Antarctica (Figure 2; note that two of the spectra were measured by our co-author Dr. Lars Lund-Hansen). Furthermore, in the Discussion section, we extensively compare our model predictions with taxonomic changes observed in polar waters by other studies, such as the shift from diatoms to *Micromonas spp* in the Canada Basin of the Arctic Ocean (lines 311-321) and the increase of cryptophytes in coastal waters of the West Antarctic Peninsula (lines 322-335).*

In our view, the key contribution of our manuscript is that it derives from first principles (from the inherent optical properties) how ice and water shape light spectra in fundamentally different ways. To our knowledge, these insights are new, they align with available data and observations, and they contribute to a better understanding of how the loss of sea ice will change the photosynthetic pigments and species composition of primary producers in polar ecosystems. For these reasons, we believe our manuscript is highly suited for Nature Communications.

I also add that the authors should address comment 3 of Rev#4 to show the extent at which phytoplankton communities can really be affected by changes in the light spectrum due to sea-ice loss.

Response: Fair point. Our analysis was thus far mostly concerned with spectra directly below the ice (Figures 2, 4 and 5), which is the prime habitat for ice algae attached to the bottom of the ice layer. However, the new Figure 3 now extends this analysis and shows how the spectra change with depth in the water column below the ice, which is relevant for phytoplankton communities in this water. The results show that, for clear ocean waters, the effect of sea ice on the spectrum in water below the ice extends to at least the 5% optical depth (16.5 m depth) (compare the spectra in Figure 3c and 3d). Hence, this new Figure shows that the loss of sea ice will alter light spectra for aquatic photosynthesis, not only for algae attached to the ice but also for phytoplankton communities in the water underneath the ice. See our above response to comment 3 for further details.

Detailed comments:

- L 61: it was discovered -> it has been shown

Response: Done (lines 60-61).

- L 115: remove 'sheet'

Response: Done (line 116).

- L 126: Change 'Now that ...' in 'Once ...'

Response: The model was validated in the preceding section. Hence, we feel that our wording ('Now that the model has been validated ...') is more accurate. We consulted a native speaker from the UK (a professor in paleo-ecology), who read this part of the text and agreed with us. Therefore, in this case, we have not adopted the reviewer's suggested wording (line 127).

- L 202 euphotic depth (zone) is defined here but it is mentioned already above in the manuscript, so this definition should appear already before

Response: Good point. We now define “euphotic zone” at first mention (on lines 82-83). Since the terms “euphotic depth” and “euphotic zone” are related but not exactly identical, we have also maintained the definition of “euphotic depth” at first mention (on lines 228-229).

- L 695: *bice* appears here for the first time, it needs to be defined in the text or at least here in the caption

Response: bice is mathematically defined as the scattering coefficient of sea ice in Eqn (4); see lines 404-414 in the Methods section. We have now also defined bice in the figure caption (line 739).

Our own changes:

1. Some sentences that were previously in the paragraphs describing the former Figure 3 (current Figure 4) have now been incorporated in the paragraphs of the new Figure 3. Therefore, the paragraphs describing the current Figure 4 have been shortened and restructured accordingly (lines 167-193).

2. We have defined optical depth based on scalar irradiance (see lines 152-153), because scalar irradiance is a more relevant measure of light availability for phytoplankton cells than planar irradiance (lines 421-423). However, although all graphs in former Figures 3 and 4 (now Figures 4 and 5) and Extended Data Figures 3-5 correctly showed scalar irradiance spectra, we had erroneously used planar instead of scalar irradiance to calculate the optical depths in these graphs. We have now corrected this error, and recalculated the optical depths based on scalar irradiance. This correction caused minor changes in several spectra and for some values in Table 1, but does not have any implications for the main conclusions of the manuscript.

*3. Very recently (on 18 March 2025), a new paper by Ahonen et al. came out in *Limnology & Oceanography* that measured irradiance spectra in 127 boreal lakes. They used the 10% optical depth (i.e., the mid-point of the euphotic zone), and confirmed the presence of three distinct spectral niches (in the green, orange and red part of the spectrum) in lakes, exactly as predicted by Stomp et al. (2007), Holtrop et al. (2021), and our current manuscript. Please note that their study only considers open water without ice cover; it does not investigate how the loss of ice will alter the spectra, so it does not interfere with our present work. However, it is a very neat study confirming the theoretically predicted existence of distinct spectral niches with actual data! We now cite the new study of Ahonen et al. (2025) as ref 28 on lines 63 and 275.*

*Ahonen, S. A., Jones, R. I., Seppälä, J., Vuorio, K. M., Tirola, M., Vähätalo, A. V. Phytoplankton absorb mainly red light in lakes with high chromophoric dissolved organic matter. *Limnology and Oceanography*, article 70034 (2025). <https://doi.org/10.1002/lno.70034>*